# *Caenorhabditis elegans* methionine/S-adenosylmethionine cycle activity is sensed and adjusted by a nuclear hormone receptor

Gabrielle E Giese[1], Melissa D Walker[1], Olga Ponomarova[1], Hefei Zhang[1], Xuhang Li[1], Gregory Minevich[2], Albertha JM Walhout[1]*

[1]Program in Systems Biology and Program in Molecular Medicine, University of Massachusetts Medical School, Worcester, United States; [2]Department of Biochemistry and Molecular Biophysics, Columbia University, New York, United States

**Abstract** Vitamin B12 is an essential micronutrient that functions in two metabolic pathways: the canonical propionate breakdown pathway and the methionine/S-adenosylmethionine (Met/SAM) cycle. In *Caenorhabditis elegans,* low vitamin B12, or genetic perturbation of the canonical propionate breakdown pathway results in propionate accumulation and the transcriptional activation of a propionate shunt pathway. This propionate-dependent mechanism requires *nhr-10* and is referred to as 'B12-mechanism-I'. Here, we report that vitamin B12 represses the expression of Met/SAM cycle genes by a propionate-independent mechanism we refer to as 'B12-mechanism-II'. This mechanism is activated by perturbations in the Met/SAM cycle, genetically or due to low dietary vitamin B12. B12-mechanism-II requires *nhr-114* to activate Met/SAM cycle gene expression, the vitamin B12 transporter, *pmp-5*, and adjust influx and efflux of the cycle by activating *msra-1* and repressing *cbs-1*, respectively. Taken together, Met/SAM cycle activity is sensed and transcriptionally adjusted to be in a tight metabolic regime.

**\*For correspondence:**
marian.walhout@umassmed.edu

**Competing interests:** The authors declare that no competing interests exist.

## Introduction

Metabolism lies at the heart of most cellular and organismal processes. Anabolic metabolism produces biomass during development, growth, cell turnover, and wound healing, while catabolic processes degrade nutrients to generate energy and metabolic building blocks. Animals must be able to regulate their metabolism in response to nutrient availability and to meet growth and energy demands. Metabolism can be regulated by different mechanisms, including the allosteric modulation of metabolic enzyme activity and the transcriptional regulation of metabolic genes. Changes in metabolic enzyme level and/or activity can result in changes in metabolic flux, which is defined as the turnover rate of metabolites through enzymatically controlled pathways. Metabolic flux can result in the accumulation or depletion of metabolites (*Watson et al., 2015*; *van der Knaap and Verrijzer, 2016*; *Wang and Lei, 2018*). These metabolites may interact with enzymes directly to allosterically affect the enzyme's catalytic properties. Alternatively, metabolites can alter the transcriptional regulation of metabolic enzymes by interacting with transcription factors (TFs) and changing their activity or localization (*Desvergne et al., 2006*; *Giese et al., 2019*). A classic example of the transcriptional regulation of metabolism is the activation of cholesterol biosynthesis genes in mammals by SREBP that responds to low levels of cholesterol (*Espenshade, 2006*).

Vitamin B12 is an essential cofactor for two metabolic enzymes: methylmalonyl-CoA mutase and methionine synthase. Methylmalonyl-coenzyme A mutase (EC 5.4.99.2) catalyzes the third step in the

breakdown of the short-chain fatty acid propionate, while methionine synthase (EC 2.1.1.13) converts homocysteine into methionine in the Methionine/S-adenosylmethionine (Met/SAM) cycle (*Figure 1A*). The Met/SAM cycle is part of one-carbon metabolism, which also includes folate metabolism and parts of purine and thymine biosynthesis (*Ducker and Rabinowitz, 2017*). The one-carbon cycle produces many important building blocks for cellular growth and repair, including nucleotides and SAM, the major methyl donor of the cell. SAM is critical for the synthesis of phosphatidylcholine, an important component of cellular membranes, as well as for the methylation of DNA, RNA, and histones (*Ye et al., 2017*). Both vitamin B12-dependent metabolic pathways have been well studied at the biochemical level; however, little is known about how these pathways are regulated transcriptionally.

The nematode *Caenorhabditis elegans* is a highly tractable model for studying the relationships between diet, disease, and metabolism (*Yilmaz and Walhout, 2014*; *Zhang et al., 2017*). *C. elegans* is a bacterivore that can thrive on both high and low vitamin B12 diets (*MacNeil et al., 2013*; *Watson et al., 2013*; *Watson et al., 2014*). We previously discovered that perturbations of the canonical propionate breakdown pathway, either genetically or by low dietary vitamin B12, results in the transcriptional activation of five genes comprising an alternative propionate breakdown pathway, or propionate shunt (*Figure 1A*, *Watson et al., 2016*). Activation of the propionate shunt occurs only with sustained propionate accumulation and absolutely depends on the nuclear hormone receptor (NHR) *nhr-10* (*Bulcha et al., 2019*). *nhr-10* functions together with *nhr-68* in a type one coherent feedforward loop known as a persistence detector (*Bulcha et al., 2019*). We refer to the regulation of gene expression by accumulation of propionate due to low vitamin B12 dietary conditions as 'B12-mechanism-I'.

Here, we report that vitamin B12 represses the expression of Met/SAM cycle genes by a propionate-independent mechanism we refer to as 'B12-mechanism-II'. We find that B12-mechanism-II is activated upon perturbation of the Met/SAM cycle, either genetically or nutritionally, due to low dietary vitamin B12. This mechanism requires another NHR, *nhr-114*, which responds to low levels of SAM. B12-mechanism-II not only activates Met/SAM cycle gene expression, it also activates the expression of the vitamin B12 transporter *pmp-5* and the methionine sulfoxide reductase *msra-1*, and represses the expression of the cystathionine beta synthase *cbs-1.* The regulation of the latter two genes increases influx and reduces efflux of the Met/SAM cycle, respectively. These findings indicate that low Met/SAM cycle activity is sensed and transcriptionally adjusted to be maintained in a tightly controlled regime. Taken together, in *C. elegans* the genetic or nutritional perturbation of the two vitamin B12-dependent pathways is sensed by two transcriptional mechanisms via different NHRs. These mechanisms likely provide the animal with metabolic adaptation to develop and thrive on different bacterial diets in the wild.

## Results

### Low dietary vitamin B12 activates two transcriptional mechanisms

As in humans, vitamin B12 acts as a cofactor in two *C. elegans* pathways: the canonical propionate breakdown pathway and the Met/SAM cycle, which is part of one-carbon metabolism (*Figure 1A*). These pathways are connected because homocysteine can be converted into cystathionine by the cystathionine beta synthase CBS-1, which after conversion into alpha-ketobutyrate is converted into propionyl-CoA. When flux through the canonical propionate breakdown pathway is perturbed, either genetically or nutritionally, that is, when dietary vitamin B12 is low, a set of genes comprising an alternative propionate breakdown pathway or propionate shunt is transcriptionally activated (*Macneil and Walhout, 2013*; *Watson et al., 2013*; *Watson et al., 2014*; *Watson et al., 2016*). Low vitamin B12 results in an accumulation of propionate, which, when sustained, activates a gene regulatory network circuit known as a type one coherent feedforward loop with an AND-logic gate composed of two transcription factors, *nhr-10* and *nhr-68* (*Bulcha et al., 2019*). The first gene in the propionate shunt, *acdh-1*, acts as a control point: its expression is induced several hundred fold when vitamin B12 is limiting (*Macneil and Walhout, 2013*; *Watson et al., 2013*; *Watson et al., 2014*; *Watson et al., 2016*).

We have previously used transgenic animals expressing the green fluorescent protein (GFP) under the control of the *acdh-1* promoter as a vitamin B12 sensor (*Arda et al., 2010*; *MacNeil et al.,*

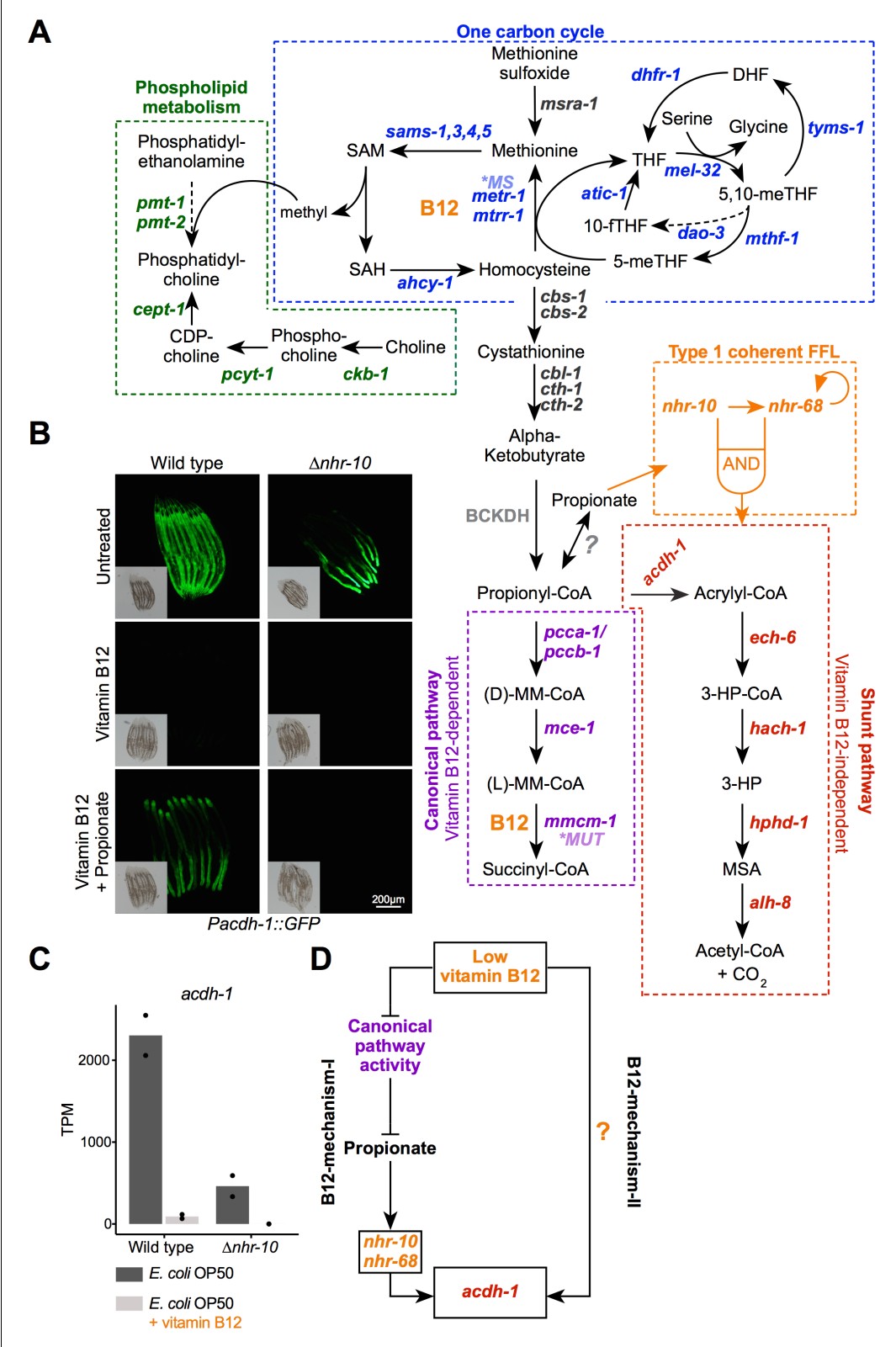

**Figure 1.** Two mechanisms of gene regulation by low vitamin B12 dietary conditions. (**A**) Cartoon of vitamin B12-related metabolic pathways in *C. elegans*. CDP –cytidine 5'-diphosphocholine; DHF – dihydrofolate; 3-HP – 3-hydroxypropionate; 5,10-meTHF – 5,10-methylenetetrahydrofolate; 5-meTHF – 5-methyltetrahydrofolate; 10-fTFH –10-formyltetrahydrofolate; BCKDH – branched-chain α-ketoacid dehydrogenase complex; MM-CoA – methylmalonyl-coenzyme A; *MUT – human methylmalonyl-coenzyme A mutase; *MS – human methionine synthase; MSA – malonic semialdehyde; *Figure 1 continued on next page*

*Figure 1 continued*

SAH – S-adenosylhomocysteine; SAM – S-adenosylmethionine; THF – tetrahydrofolate; FFL – feed forward loop. Dashed arrows indicate multiple reaction steps. (B) Fluorescence microscopy images of *Pacdh-1::GFP* reporter animals in wild type and Δ*nhr-10* mutant background with different supplements as indicated. Insets show brightfield images. (C) RNA-seq data of *acdh-1* mRNA with and without 20 nM vitamin B12 in wild type and Δ*nhr-10* mutant animals (*Bulcha et al., 2019*). Datapoints show each biological replicate and the bar represents the mean. TPM – transcripts per million. p adjusted values are provided in *Supplementary file 1*. (D) Cartoon illustrating two mechanisms of gene regulation by low vitamin B12 dietary conditions.

The online version of this article includes the following figure supplement(s) for figure 1:

**Figure supplement 1.** Fluorescent microscopy images of *Pacdh-1::GFP* animals in wild type and Δ*nhr-10* mutant backgrounds with supplemented metabolites as indicated.

**Figure supplement 2.** Boxplot showing median and interquartile range of normalized GFP intensity measurements of fluorescent images shown in *Figure 1B*.

---

*2013*; *Watson et al., 2014*). In these *Pacdh-1::GFP* animals, GFP expression is high throughout the intestine on an *E. coli* OP50 diet, which is low in vitamin B12, and GFP expression is very low on a *Comamonas aquatica* DA1877 diet that is high in vitamin B12 (*MacNeil et al., 2013*; *Watson et al., 2014*). Low GFP expression resulting from vitamin B12 supplementation to the *E. coli* OP50 diet can be overcome by addition of propionate (*Figure 1B*, *Figure 1—figure supplements 1* and *2*; *Bulcha et al., 2019*). The activation of *acdh-1* expression in response to accumulating propionate is completely dependent on *nhr-10* (*Figure 1B*, *Figure 1—figure supplements 1* and *2*; *Bulcha et al., 2019*). Interestingly, we found that while GFP levels are reduced in the anterior intestine, there is still remaining GFP expression in the posterior intestine in *Pacdh-1::GFP* transgenic animals lacking *nhr-10* (*Figure 1B*, *Figure 1—figure supplements 1* and *2*). Since *nhr-10* is absolutely required to mediate the activation of *acdh-1* by propionate, this means that there is another, propionate-independent mechanism of activation. Importantly, the residual GFP expression in *Pacdh-1::GFP*; Δ*nhr-10* was completely repressed by the supplementation of vitamin B12 (*Figure 1B*, *Figure 1—figure supplements 1* and *2*). This result was confirmed by inspecting our previously published RNA-seq data: in Δ*nhr-10* animals there is residual endogenous *acdh-1* expression which is eliminated by the addition of vitamin B12 (*Figure 1C*, *Supplementary file 1*; *Bulcha et al., 2019*). These results demonstrate that there is another mechanism by which low vitamin B12 activates gene expression that is independent of propionate accumulation, which occurs when flux through the canonical propionate breakdown pathway is perturbed. We refer to the activation of gene expression in response to canonical propionate breakdown perturbation as 'B12-mechanism-I' and the other, propionate-independent mechanism as 'B12-mechanism-II' (*Figure 1D*).

## Met/SAM cycle perturbations activate B12-mechanism-II

To determine the mechanism by which 'B12-mechanism-II' is activated, we used the *Pacdh-1::GFP* vitamin B12 sensor in the Δ*nhr-10* mutant background, which cannot respond to B12-mechanism-I. We first performed a forward genetic screen using ethyl methanesulfonate (EMS) to find mutations that activate GFP expression in *Pacdh-1::GFP*;Δ*nhr-10* animals in the presence of vitamin B12 (*Figure 2A*). We screened ~8000 genomes and identified 27 mutants, 16 of which were viable and produced GFP-expressing offspring. Seven of these mutants were backcrossed with the *Pacdh-1::GFP*;Δ*nhr-10* parent strain. Single-nucleotide polymorphism mapping and whole genome sequencing revealed mutations in *metr-1*, *mtrr-1*, *sams-1*, *mthf-1*, and *pmp-5* (*Figure 2B*). The first four genes encode enzymes that function directly in the Met/SAM cycle (*Figure 1A*). *metr-1* is the single ortholog of human methionine synthase; *mtrr-1* is the ortholog of *MTRR* that encodes methionine synthase reductase; *sams-1* is orthologous to human *MAT1A* and encodes a SAM synthase; and *mthf-1* is the ortholog of human *MTHFR* that encodes methylenetetrahydrofolate reductase. We also found mutations in *pmp-5*, an ortholog of human *ABCD4*, which encodes a vitamin B12 transporter (*Coelho et al., 2012*).

Next, we performed a reverse genetic RNAi screen using a library of predicted metabolic genes in *Pacdh-1::GFP*;Δ*nhr-10* animals fed *E. coli* HT115 bacteria (the bacterial diet used for RNAi experiments) supplemented with vitamin B12 (*Figure 2C*). In these animals, GFP expression is off and we looked for those RNAi knockdowns that activated GFP expression in the presence of vitamin B12. Out of more than 1400 genes tested, RNAi of only five genes resulted in activation of GFP

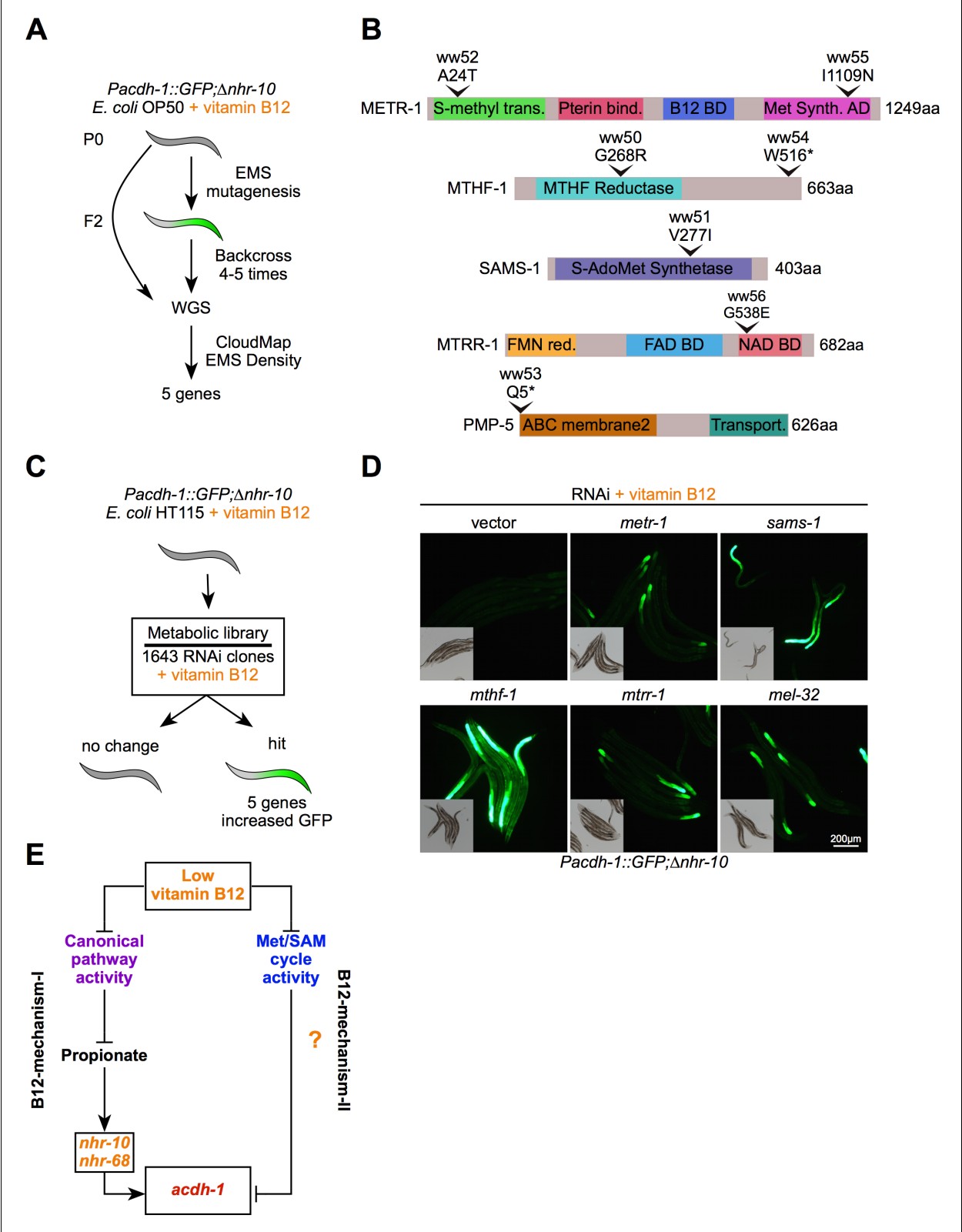

**Figure 2.** Met/SAM cycle perturbations activate B12-mechanism-II. (**A**) Workflow for the EMS mutagenesis screen using *Pacdh-1::GFP;Δnhr-10* reporter animals supplemented with 20 nM vitamin B12. WGS – whole genome sequencing. (**B**) To-scale cartoons of amino acid changes in the proteins encoded by the genes found in the forward genetic screen. (**C**) Workflow for RNAi screen using *Pacdh-1::GFP;Δnhr-10* reporter animals supplemented with 20 nM vitamin B12. (**D**) Fluorescence microscopy images of *Pacdh-1::GFP;Δnhr-10* reporter animals subjected to RNAi of the indicated metabolic

*Figure 2 continued on next page*

*Figure 2 continued*
genes. Insets show brightfield images. (E) Cartoon illustrating the activation of B12-mechanism-II by low vitamin B12 or genetic perturbations in the Met/SAM cycle.

expression: *metr-1*, *mtrr-1*, *sams-1*, *mthf-1*, and *mel-32* (*Figure 2D*). Four of these genes were also found in the forward genetic screen (*Figure 2B*). The fifth gene, *mel-32,* also functions in one-carbon metabolism (*Figure 1A*). It is an ortholog of human *SHMT1* and encodes serine hydroxymethyltrans-ferase that converts serine into glycine thereby producing 5,10-methylenetetrahydrofolate (5,10-meTHF), the precursor of 5-meTHF, which donates a methyl group in the reaction catalyzed by METR-1 (*Figure 1A*). These results show that genetic perturbations in Met/SAM cycle genes activate B12-mechanism-II, even in the presence of vitamin B12. This indicates that reduced activity of the Met/SAM cycle, either due to genetic perturbations or as a result of low dietary vitamin B12 acti-vates B12-mechanism-II (*Figure 2E*). Therefore, genetic perturbations in either pathway that uses vitamin B12 as a cofactor activate the vitamin B12 sensor.

## B12-mechanism-II activates Met/SAM cycle gene expression in response to Met/SAM cycle perturbations

To ask what other genes are activated by B12-mechanism-II, we performed RNA-seq using four Met/SAM cycle mutants identified in the forward genetic screen. All strains were supplemented with vita-min B12 and gene expression profiles of the Met/SAM cycle mutants were compared to the parental *Pacdh-1::GFP;Δnhr-10* strain. We found a set of 110 genes that are upregulated in all four Met/SAM cycle mutants, and a smaller set of 11 genes that are downregulated (*Figure 3A*, *Supplementary file 2*). Importantly, endogenous *acdh-1* was upregulated in each of the Met/SAM cycle mutants, validating the results obtained with the *Pacdh-1::GFP* vitamin B12 sensor (*Supplementary file 2*). Remarkably, we found that most Met/SAM cycle genes are significantly upregulated in each of the Met/SAM cycle mutants, and one of them, *mel-32*, was significantly increased in three of the four mutants (the fourth mutant just missing the selected statistical thresh-old) (*Figure 3B*, *Supplementary file 2*).

The finding that Met/SAM cycle genes are transcriptionally activated in response to genetic Met/SAM cycle perturbations implies that these genes may also be activated by low vitamin B12. Indeed, inspection of our previously published RNA-seq data (*Bulcha et al., 2019*) revealed that expression of Met/SAM cycle genes is repressed by vitamin B12 (*Figure 3C*, *Supplementary file 1*). In contrast to propionate shunt genes, however, Met/SAM cycle genes are not induced in response to propio-nate supplementation, nor are these genes affected in *nhr-10* or *nhr-68* mutants, which are the medi-ators of the propionate response (B12-mechanism-I, *Figure 3D*, *Supplementary file 1*). Therefore, Met/SAM cycle gene expression is activated by B12-mechanism-II in response to either genetic or nutritional (low vitamin B12) perturbations in the Met/SAM cycle (*Figure 3E*).

## nhr-114 mediates Met/SAM cycle activation in response to low Met/SAM cycle flux

How do perturbations in the Met/SAM cycle activate B12-mechanism-II? There are two components to this question: (1) which TF(s), and (2) which metabolite(s) mediate Met/SAM cycle gene induction? We first focused on identifying the TF(s) involved in B12-mechanism-II. Previously, we identified more than 40 TFs that activate the *Pacdh-1::GFP* vitamin B12 sensor in wild type animals (*MacNeil et al., 2015*). Subsequently, we found that only a subset of these TFs are involved in B12-mechanism-I, most specifically *nhr-10* and *nhr-68* (*Bulcha et al., 2019*). To identify TFs involved in B12-mechanism-II, we performed RNAi of all TFs that regulate the *acdh-1* promoter (*MacNeil et al., 2015*; *Bulcha et al., 2019*) in *Pacdh-1::GFP;Δnhr-10* animals harboring mutations in Met/SAM genes. As mentioned above, these animals express moderate levels of GFP in response to the activation of B12-mechanism-II by Met/SAM cycle mutations. RNAi of several TFs reduced GFP expression, including *elt-2*, *nhr-23*, *cdc-5L*, *lin-26*, *sbp-1*, and *nhr-114* (*Figure 4A*, *Figure 4—figure supplement 1*). Most of these TFs function at a high level in the intestinal gene regulatory network, elicit gross physiological phenotypes when knocked down, and are also involved in B12-mechanism-I (*MacNeil et al., 2015*; *Bulcha et al., 2019*). For instance, *elt-2* is a master regulator that is required

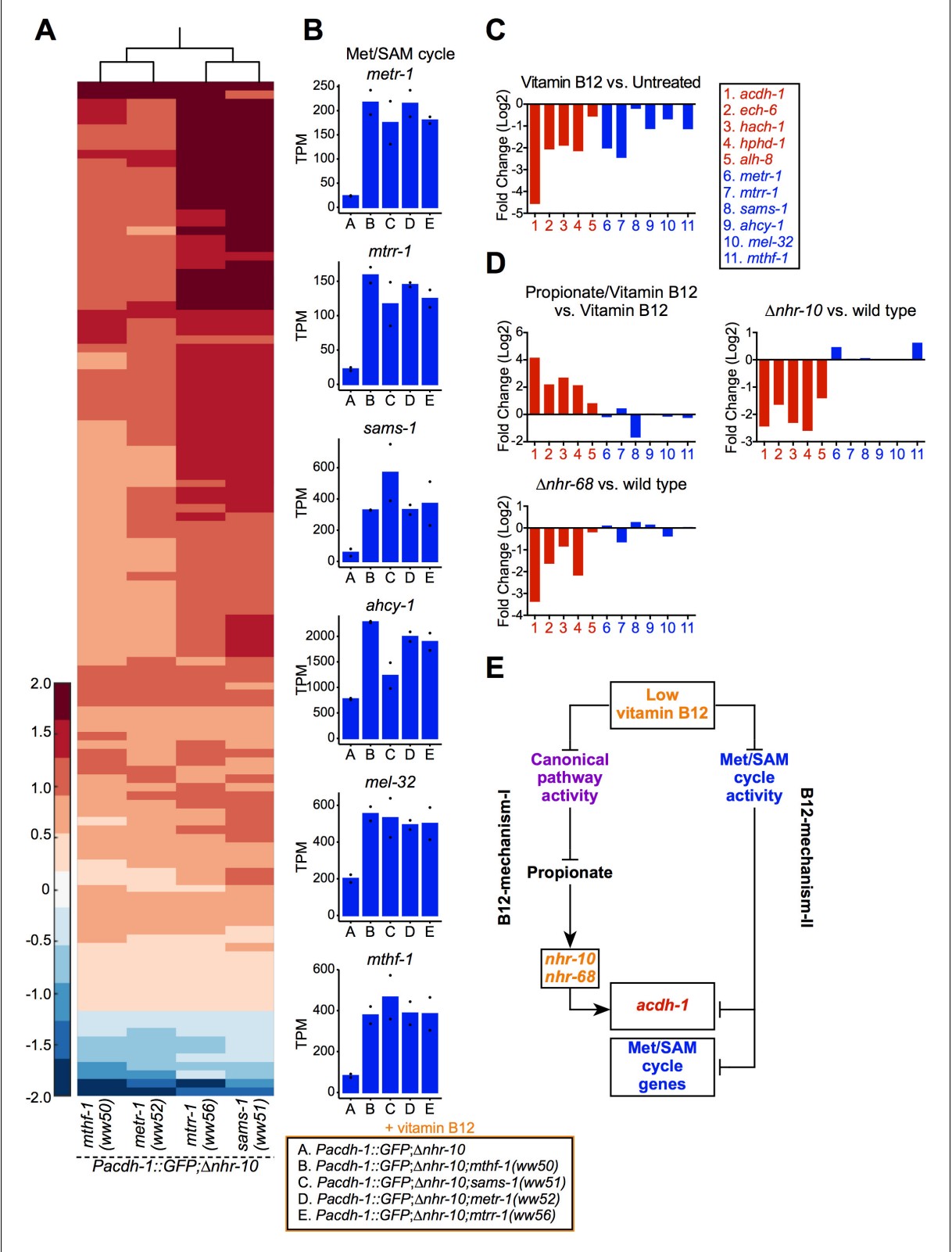

**Figure 3.** Genetic mutations in Met/SAM cycle genes activate Met/SAM cycle gene expression. (A) Hierarchal clustering of log10-transformed fold change RNA-seq data. Changes are relative to the *Pacdh-1::GFP;Δnhr-10* parent strain with the cutoffs of a fold change of 1.5 and a p adjusted value less than 0.01. Only genes that change in expression in all four Met/SAM cycle gene mutants are shown. (B) Bar graphs of Met/SAM cycle gene expression by RNA-seq in the four Met/SAM cycle mutants and in the *Pacdh-1::GFP;Δnhr-10* parent strain. Datapoints show each biological replicate

*Figure 3 continued on next page*

Figure 3 continued

and the bar represents the mean. TPM – transcripts per million. p adjusted values are provided in *Supplementary file 2*. (C) RNA-seq comparison of Met/SAM cycle (blue) and propionate shunt (red) gene expression in response to 20 nM vitamin B12. Bar represents the mean of two biological replicates. p adjusted values are provided in *Supplementary file 1*. (D) Comparison of Met/SAM cycle and propionate shunt genes in response to 20 nM vitamin B12 plus 40 mM propionate or in *nhr-10* and *nhr-68* mutant animals. Bar represents the mean of two biological replicates. p adjusted values are provided in *Supplementary file 1*. (E) Cartoon illustrating met/SAM cycle gene activation in response to B12-mechanism-II.

for the intestinal expression of most if not all genes and is therefore not specific (*McGhee et al., 2009*; *MacNeil et al., 2015*). The TFs *nhr-23* and *lin-26* were not found to be involved in B12-mechanism-I and could therefore potentially be involved in B12-mechanism-II. However, RNAi of these TFs causes severe developmental delay across all strains and thus their contribution could be less specific (*Figure 4—figure supplement 1*). Only *nhr-114* RNAi, which we previously found not to be involved in B12-mechanism-I, specifically repressed GFP expression in the *nhr-10* deletion mutant background (*Figure 4A*, *Figure 4—figure supplements 2* and *3*; *Bulcha et al., 2019*). This indicates that *nhr-114* mediates the response to B12-mechanism-II. Interestingly, *nhr-114* RNAi greatly slowed development in Met/SAM cycle mutants but not in the parental strain (*Figure 4A*, inset, *Figure 4—figure supplement 2*, insets). This indicates that combined *nhr-114* and Met/SAM cycle perturbations produce a synthetic sick phenotype and points to the functional importance of *nhr-114* in Met/SAM cycle metabolism.

Previously, it has been reported that *nhr-114* loss-of-function mutants develop slowly and are sterile when fed a diet of *E. coli* OP50 bacteria, while they grow faster and are fertile on a diet of *E. coli* HT115 (*Gracida and Eckmann, 2013*). *E. coli* HT115 bacteria are thought to contain higher levels of vitamin B12 than *E. coli* OP50 cells (*Watson et al., 2014*; *Revtovich et al., 2019*). Therefore, we asked whether Δ*nhr-114* mutant phenotypes could be rescued by vitamin B12 supplementation. Indeed, we found that both sterility and slow development of Δ*nhr-114* mutants fed *E. coli* OP50 could be rescued by supplementation of vitamin B12 (*Figure 4B and C*, *Figure 4—figure supplement 4*).

The Met/SAM cycle generates SAM, the major methyl donor of the cell that is critical for the synthesis of phosphatidylcholine (*Ye et al., 2017*; *Figure 1A*). It has previously been shown that perturbation of *sams-1*, which converts methionine into SAM (*Figure 1A*), leads to a strong reduction in fecundity and large changes in gene expression (*Li et al., 2011*; *Ding et al., 2015*). Importantly, these phenotypes can be rescued by supplementation of choline, which supports an alternative route to phosphatidylcholine biosynthesis (*Walker et al., 2011*; *Figure 1A*). Therefore, in *C. elegans*, the primary biological function of the Met/SAM cycle is to produce methyl donors that facilitate the synthesis of phosphatidylcholine. We found that the Δ*nhr-114* mutant phenotypes can also be rescued by either methionine or choline supplementation (*Figure 4B and C*, *Figure 4—figure supplement 4*). Although Gracida and Eckmann did not find that methionine rescued *nhr-114* RNAi knock down animals, it is possible this was due to methodological differences. For example, they added bulk L-amino acids and at only one concentration. The actual concentration of ingested methionine might have been masked by other amino acids or was simply too low to provide an observable rescue (*Gracida and Eckmann, 2013*). Rescue of *nhr-114* mutant phenotypes by choline and methionine support the hypothesis that low levels of phosphatidylcholine are the underlying cause of the Δ*nhr-114* mutant phenotypes on low vitamin B12 diets, when B12-mechanism-II cannot activate Met/SAM cycle gene expression. This result provides further support for the functional involvement of *nhr-114* in the Met/SAM cycle, and leads to the prediction that *nhr-114* activates Met/SAM cycle gene expression in response to genetic or nutritional perturbation of the cycle's activity. To directly test this prediction, we performed RNA-seq on *Pacdh-1::GFP* and *Pacdh-1::GFP;Δnhr-10;metr-1 (ww52)* animals supplemented with vitamin B12 and subjected to *nhr-114* or vector control RNAi. We also included *Pacdh-1::GFP;Δmetr-1* animals, which have wild type *nhr-10*. We found that Met/SAM cycle genes are robustly induced in both *metr-1* mutants, recapitulating our earlier observation (*Figures 4D* and *3B*, *Figure 4—figure supplement 5*, *Supplementary files 3* and *2*). Importantly, we found that this induction is absolutely dependent on *nhr-114* (*Figure 4D*, *Figure 4—figure supplement 5*, *Supplementary file 3*). Overall, the induction of 32 of the 110 genes that are upregulated by Met/SAM cycle perturbations requires *nhr-114* (*Figure 4E*). These genes are candidate modulators of Met/SAM cycle function. Interestingly, in the presence of vitamin B12, basal Met/SAM

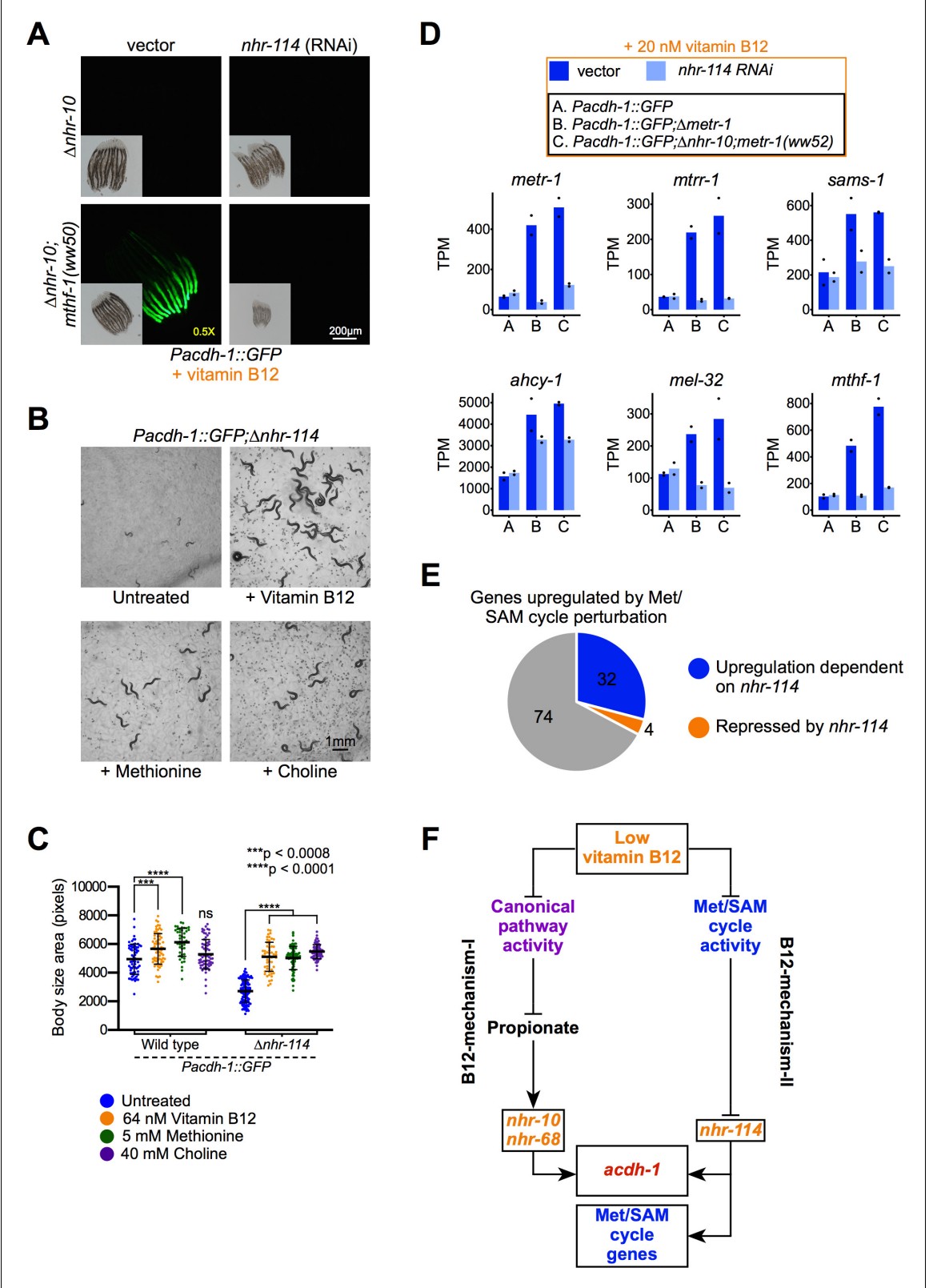

**Figure 4.** *nhr-114* is required for B12-mechanism-II in response to Met/SAM cycle perturbations. (**A**) *nhr-114* RNAi reduces GFP expression in *Pacdh-1::GFP;Δnhr-10* animals harboring Met/SAM cycle gene mutations. Insets show brightfield images. (**B**) Δ*nhr-114* mutant growth and fertility phenotypes are rescued by vitamin B12, methionine or choline supplementation. Difference in exposure time is indicated in yellow. (**C**) Quantification of body size of wild type and Δ*nhr-114* mutant animals that are untreated or supplemented with either vitamin B12, methionine, or choline. Trial two is shown in

*Figure 4 continued on next page*

*Figure 4 continued*

**Figure 4—figure supplement 4.** Statistical significance was determined by the Kruskal-Wallis test with post hoc comparison using Dunn's multiple comparison test. (D) Bar graphs of Met/SAM cycle gene expression by RNA-seq in *Pacdh-1::GFP*, *Pacdh-1::GFP;Δmetr-1* and *Pacdh-1::GFP;Δnhr-10; metr-1(ww52)* animals treated with vector control or *nhr-114* RNAi. Datapoints show each biological replicate and the bar represents the mean. TPM – transcripts per million. p adjusted values are provided in *Supplementary file 3*. (E) Pie chart showing portion of genes upregulated by Met/SAM cycle perturbation that are *nhr-114* dependent. (F) Cartoon illustrating the requirement of *nhr-114* in B12-mechanism-II.

The online version of this article includes the following figure supplement(s) for figure 4:

**Figure supplement 1.** Diluted RNAi of indicated TFs repress *Pacdh-1::GFP* in met/SAM cycle mutants harboring a deletion in *nhr-10*.

**Figure supplement 2.** *nhr-114* RNAi represses *Pacdh-1::GFP* expression in all Met/SAM cycle mutants.

**Figure supplement 3.** Boxplot showing median and interquartile range of normalized GFP intensity measurements of fluorescent images shown in *Figure 4A*.

**Figure supplement 4.** Replicate experiment of rescue of *Δnhr-114* mutant developmental phenotype by vitamin B12, methionine, and choline shown in *Figure 4C*.

**Figure supplement 5.** Relative mRNA level expression (fold change) as determined by qRT-PCR of Met/SAM cycle genes whose p adjusted values were high in RNA-seq shown in *Figure 4D*.

cycle gene expression is not affected by *nhr-114* RNAi (*Figure 4D*, *Figure 4—figure supplement 5*, *Supplementary file 3*). Together with the observation that *nhr-114* and Met/SAM cycle perturbations produce a synthetic sick phenotype (*Figure 4A*, *Figure 4—figure supplement 2*), this indicates that *nhr-114* is specifically involved in B12-mechanism-II, which is activated when the activity of the cycle is hampered. Taken together, perturbations in the Met/SAM cycle elicited either by low dietary vitamin B12 or by genetic perturbations activate Met/SAM cycle gene expression by B12-mechanism-II, which requires the function of *nhr-114* (*Figure 4F*).

## Methionine and choline supplementation suppress B12-Mechanism-II

Which metabolites are involved in the activation of B12-mechanism-II? To start addressing this question, we first performed targeted metabolomics by gas chromatography-mass spectrometry (GC-MS) on animals fed a vitamin B12-deplete *E. coli* OP50 diet with or without supplementation of vitamin B12. The enhanced activity of the Met/SAM cycle in the presence of supplemented vitamin B12 is apparent because methionine levels increase, while homocysteine levels decrease (*Figure 5A*). 3-hydroxypropionate levels are also dramatically decreased by vitamin B12 supplementation, because propionate is preferentially degraded by the canonical propionate breakdown pathway (*Figure 5A*; *Watson et al., 2016*). We reasoned that either low methionine, low SAM, low phosphatidylcholine or high homocysteine could activate B12-mechanism-II when activity of the Met/SAM cycle is perturbed.

We first tested the possibility that the accumulation of homocysteine in Met/SAM cycle mutants may activate B12-mechanism-II. In *C. elegans*, RNAi of *cbs-1* causes the accumulation of homocysteine (*Vozdek et al., 2012*). Therefore, we reasoned that, if homocysteine accumulation activates B12-mechanism-II, RNAi of *cbs-1*, should increase GFP expression in the *Pacdh-1::GFP* vitamin B12 sensor. Remarkably, however, we found the opposite: RNAi of *cbs-1* repressed GFP expression in *Pacdh-1::GFP;Δnhr-10* animals but not in the Met/SAM cycle mutants (*Figure 5B*, *Figure 5—figure supplements 1* and *2*). This indicates that a build-up of homocysteine is not the metabolic mechanism that activates B12-mechanism-II. The repression of GFP expression by *cbs-1* RNAi in *Pacdh-1::GFP;Δnhr-10* animals could be explained by a decrease in the conversion of homocysteine into cystathionine and an increase in the conversion into methionine resulting in support of Met/SAM cycle activity (*Figure 1A*). In sum, B12-mechanism-II is not activated by a build-up of homocysteine.

Next, we explored whether low methionine, low SAM or low phosphatidylcholine activates B12-mechanism-II. We found that either methionine or choline supplementation dramatically repressed GFP expression in *Pacdh-1::GFP;Δnhr-10* animals (*Figure 5C*, *Figure 5—figure supplement 3*). However, neither metabolite greatly affected GFP levels in wild type reporter animals (*Figure 5C*, *Figure 5—figure supplement 3*). Since these animals are fed vitamin B12-depleted *E. coli* OP50 bacteria and have functional *nhr-10* and *nhr-68* TFs, GFP expression is likely high due to propionate accumulation, that is, B12-mechanism-I. Importantly, either methionine or choline supplementation also repressed GFP expression induced by B12-mechanism-II due to mutations in *metr-1* or *mthf-1* (*Figure 5D*, *Figure 5—figure supplement 3*). However, while choline supplementation repressed

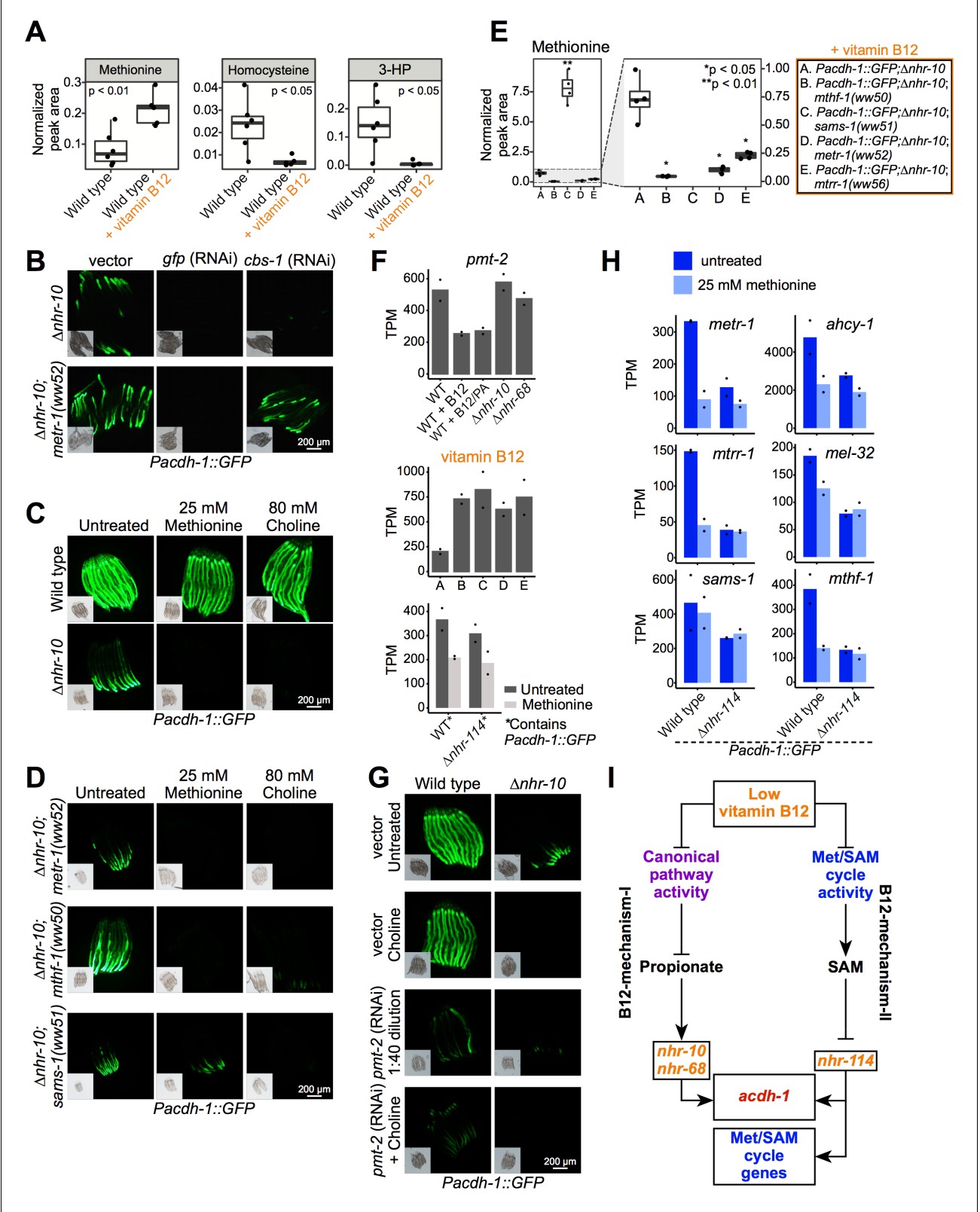

**Figure 5.** Methionine and choline supplementation suppress vitamin B12-mechanism-II. (**A**) Box plots showing GC-MS data from wild type animals fed *E. coli* OP50 with or without supplemented 64 nM vitamin B12. Statistical significance determined by two-tailed t-test. 3-HP – 3-hydroxypropionate. (**B**) *cbs-1* RNAi reduces GFP expression in *Pacdh-1::GFP;Δnhr-10* and *Pacdh-1::GFP;Δnhr-10* mutants harboring Met/SAM cycle gene mutations. (**C**) Methionine or choline supplementation represses GFP expression in *Pacdh-1::GFP;Δnhr-10* animals. Insets show brightfield images. (**D**) Methionine or

*Figure 5 continued on next page*

*Figure 5 continued*

choline supplementation represses GFP expression in *Pacdh-1::GFP;Δnhr-10;metr-1(ww52)* and *Pacdh-1::GFP;Δnhr-10;mthf-1(ww50)* animals, while choline but not methionine supplementation represses GFP expression in *Pacdh-1::GFP;Δnhr-10;sams-1(ww51)* animals. This experiment was performed without vitamin B12 supplementation. (E) GC-MS quantification of methionine levels in Met/SAM cycle mutants and in the parental strain. Statistical significance was determined using one-way ANOVA with post-hoc comparison using Dunnett's T3 test. (F) RNA-seq data of *pmt-2* expression in each of the datasets. PA – propionic acid; X axis labels match legend in panel E. Datapoints show each biological replicate and the bar represents the mean. TPM – transcripts per million. p adjusted values are provided in *Supplementary files 1, 2* and *4*. (G) *pmt-2* RNAi suppresses GPF expression in *Pacdh-1::GFP;Δnhr-10* animals. *pmt-2* RNAi experiments were diluted with vector control RNAi to circumvent strong deleterious phenotypes. (H) Bar graphs of RNA-seq data showing Met/SAM cycle gene expression in wild type and Δ*nhr-114* animals with and without methionine supplementation. p adjusted values are provided in *Supplementary file 4*. (I) Cartoon illustrating B12-mechanism-II whereby low vitamin B12 reduces Met/SAM cycle activity, leading to the depletion of SAM and the activation of met/SAM cycle gene expression mediated by *nhr-114*.

The online version of this article includes the following figure supplement(s) for figure 5:

**Figure supplement 1.** Met/SAM cycle point mutants expressing *Pacdh-1::GFP* treated with *cbs-1* RNAi.
**Figure supplement 2.** Boxplot showing median and interquartile range of normalized GFP intensity measurements of fluorescent images shown in *Figure 5B*.
**Figure supplement 3.** Boxplot showing median and interquartile range of normalized GFP intensity measurements of fluorescent images shown in *Figure 5C and D*.
**Figure supplement 4.** Relative mRNA level expression (fold change) as determined by qRT-PCR of *pmt-2* whose p adjusted value was high in RNA-seq shown in *Figure 5F*.
**Figure supplement 5.** Wild type or Δ*nhr-114* expressing *Pacdh-1::GFP* with *pmt-2* RNAi supplemented with and without choline.
**Figure supplement 6.** Boxplot showing median and interquartile range of normalized GFP intensity measurements of fluorescent images shown in *Figure 5G*.
**Figure supplement 7.** Relative mRNA level expression (fold change) as determined by qRT-PCR of Met/SAM cycle genes whose p adjusted value was high in RNA-seq shown in *Figure 5H*.Datapoints show each biological replicate and bar represents the mean.

GFP expression in *sams-1(ww51)* mutants, methionine supplementation did not (*Figure 5D*, *Figure 5—figure supplement 3*). SAMS-1 converts methionine into SAM, and methionine levels are greatly increased in *sams-1* mutant animals, while being reduced in *metr-1*, *mtrr-1*, and *mthf-1* mutants (*Figures 1A* and *5E*). Since methionine levels are elevated in *sams-1* mutants, and because methionine supplementation cannot suppress GFP expression in these mutants, these results indicate that low methionine is not the direct activator of B12-mechanism-II, but rather that it is either low SAM, or low phosphatidylcholine, both of which require methionine for their synthesis. In *metr-1* and *mthf-1* mutants methionine supplementation supports the synthesis of SAM and phosphatidylcholine, and in these mutants, methionine supplementation would therefore act indirectly. We did observe a mild reduction in GFP levels upon methionine supplementation in *sams-1(ww51)* animals likely because it is not a complete loss-of-function allele, and/or functional redundancy with three other *sams* genes (*Figure 5D*, *Figure 5—figure supplement 3*).

To distinguish between the possibilities of low SAM or low phosphatidylcholine activating B12-mechanism-II, we next focused on *pmt-2*. PMT-2 is involved in the second step of the conversion of phosphatidylethanolamine into phosphatidylcholine (*Yilmaz and Walhout, 2016*; *Figure 1A*). The expression of *pmt-2* is repressed by vitamin B12, but not activated by propionate, is not under the control of *nhr-10* or *nhr-68*, and is activated in Met/SAM cycle mutants (*Bulcha et al., 2019*; *Figure 5F*, *Figure 5—figure supplement 4*, *Supplementary files 1* and *2*). We reasoned that *pmt-2* RNAi might allow us to discriminate whether low SAM or low phosphatidylcholine activates B12-mechanism-II. Specifically, we would expect *pmt-2* RNAi to activate B12-mechanism-II if low phosphatidylcholine is the main cause, and consequently that GFP expression would increase. Due to the severe growth delay caused by RNAi of *pmt-2*, we diluted the RNAi bacteria with vector control bacteria. We found that *pmt-2* RNAi decreased GFP expression particularly in *Pacdh-1::GFP;Δnhr-10* animals (*Figure 5G*, *Figure 5—figure supplements 5* and *6*). Predictably, choline supplementation rescued the growth defect caused by *pmt-2* RNAi. While it is possible that RNAi may cause off-target effects, the rescue by choline suggests that the knock down by *pmt-2* RNAi is specific. Importantly, in both choline-supplemented and dilute RNAi conditions, *pmt-2* RNAi repressed *Pacdh-1::GFP*. Therefore, we conclude that low phosphatidylcholine is not the inducer of B12-mechanism-II. Instead, our results support a model in which low SAM levels activate B12-mechanism-II. *pmt-2* RNAi likely represses the *Pacdh-1::GFP* transgene because SAM levels increase when the methylation reaction converting phosphatidylethanolamine into phosphatidylcholine, which depends on SAM is

blocked (*Ye et al., 2017*). Taken together, our data support a model in which low Met/SAM cycle activity results in low SAM levels, which activates B12-mechanism-II.

Next, we asked whether *nhr-114* is required for the transcriptional response to low SAM. Since SAM is not stable and may not be easily absorbed by *C. elegans*, we used methionine supplementation, which supports SAM synthesis, except in *sams-1* mutant animals. We performed RNA-seq in wild type and Δ*nhr-114* mutant animals with or without methionine supplementation and found that Met/SAM cycle genes are repressed by methionine supplementation in wild type, but not Δ*nhr-114* mutant animals (*Figure 5H*, *Figure 5—figure supplement 7*, *Supplementary file 4*). Therefore, low SAM levels due to vitamin B12 depletion activate B12-mechanism-II in an *nhr-114*-dependent manner (*Figure 5I*).

## B12-mechanism-II activates influx and represses efflux of the Met/SAM cycle

Some of the most strongly regulated vitamin B12-repressed genes include *msra-1* and *pmp-5* (*Figure 6A and B*, *Supplementary file 1*; *Bulcha et al., 2019*). In the forward genetic screen, we identified a mutation in *pmp-5* that activates the *Pacdh-1::GFP* transgene in Δ*nhr-10* mutant animals in the presence of supplemented vitamin B12. As mentioned above, *pmp-5* is an ortholog of human

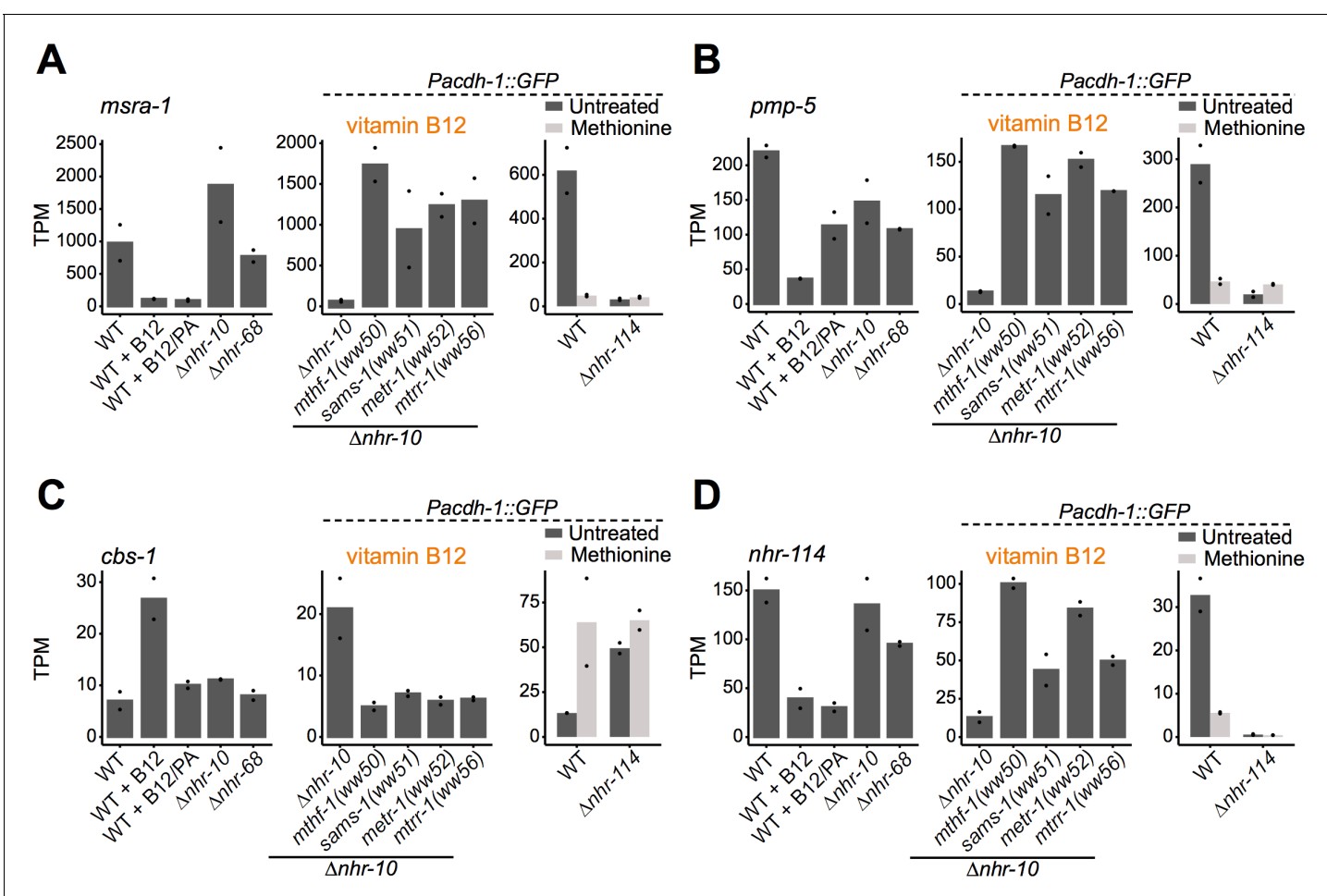

**Figure 6.** *nhr-114* transcriptionally regulates Met/SAM cycle influx and efflux. (A, B, C) RNA-seq data from each of the experiments for *msra-1* (A), *pmp-5* (B), *cbs-1* (C) and *nhr-114* (D). Datapoints show each biological replicate and the bar represents the mean. TPM – transcripts per million. p adjusted values are provided in *Supplementary files 1*, *2* and *4*.

The online version of this article includes the following figure supplement(s) for figure 6:

**Figure supplement 1.** Relative mRNA level expression (fold change) as determined by qRT-PCR of *cbs-1* whose p adjusted value was high in RNA-seq shown in *Figure 6C*.

*ABCD4*, which encodes a vitamin B12 transporter (*Coelho et al., 2012*). Thus, increased B12 transport may be used by the animal as a mechanism to increase Met/SAM cycle activity. *msra-1* encodes methionine sulfoxide reductase that reduces methionine sulfoxide to methionine (*Figure 1A*). This gene provides an entry point into the Met/SAM cycle by increasing levels of methionine. This observation prompted us to hypothesize that perturbation of Met/SAM cycle activity, either by low dietary vitamin B12 or by genetic perturbations in the cycle, may activate the expression of these genes. Indeed, both genes are induced in the Met/SAM cycle mutants (*Figure 6A and B*, *Supplementary file 2*). Further, both genes are repressed by methionine supplementation, in an *nhr-114*-dependent manner (*Figure 6A and B*, *Supplementary file 4*). As a putative transporter of vitamin B12 from the lysosome to the cytosol, *pmp-5* is also important for vitamin B12-mechanism-I. Indeed, pmp-5 is upregulated in response to propionate supplementation, and is also regulated by *nhr-10* and *nhr-68* (*Figure 6B*, *Supplementary file 1*). We also noticed that the expression level changes of *cbs-1* are opposite of those of *msra-1* and *pmp-5*: *cbs-1* is activated by vitamin B12, repressed in Met/SAM cycle mutants, and activated by methionine in an *nhr-114*-dependent manner (*Figure 6C*, *Figure 6—figure supplement 1*, *Supplementary files 1*, *2* and *4*). As mentioned above, *cbs-1* encodes cystathionine beta synthase, which converts homocysteine into cystathionine (*Figure 1A*). Reduced *cbs-1* expression upon Met/SAM cycle perturbations would therefore likely prevent carbon efflux. Finally, *nhr-114* expression itself is repressed by both vitamin B12 and methionine and activated by perturbations in Met/SAM cycle genes (*Figure 6D*, *Supplementary files 1*, *2* and *4*). This suggests that *nhr-114* activates its own expression, similarly as the auto-activation of *nhr-68* in response to propionate accumulation (*Bulcha et al., 2019*). *nhr-114* expression is not under the control of B12-mechanism-I because it does not change when propionate is supplemented or when *nhr-10* is deleted. However, *nhr-114* is mildly repressed in Δ*nhr-68* mutants. This suggests that there may be some crosstalk between the two B12 mechanisms (*Figure 6D*; *Bulcha et al., 2019*). Taken together, B12-mechanism-II is employed when Met/SAM cycle activity is perturbed to increase Met/SAM cycle gene expression as well as Met/SAM cycle activity and influx, and to decrease Met/SAM cycle efflux (*Figure 7*).

## Discussion

We have discovered a second mechanism by which vitamin B12 regulates gene expression in *C. elegans*. This B12-mechanism-II is different from B12-mechanism-I, which we previously reported to

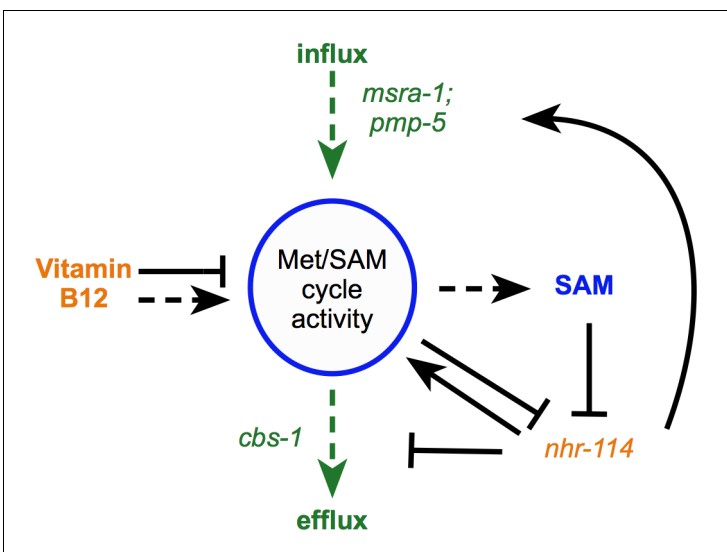

**Figure 7.** Model of B12-mechanism-II. Vitamin B12 modulates the transcription of Met/SAM cycle genes and controls in/efflux through the sensing of SAM by *nhr-114*. Dashed arrows indicate regulation of metabolic activity. Solid arrows indicate transcriptional regulation. Vitamin B12 increases Met/SAM cycle metabolic activity producing SAM, which represses *nhr-114* transcription thereby reducing the expression of Met/SAM cycle related genes.

transcriptionally activate a propionate shunt in response to persistent accumulation of this short-chain fatty acid (*Bulcha et al., 2019*). B12-mechanism-I is activated under low dietary vitamin B12 conditions, or when the canonical vitamin B12-dependent propionate breakdown pathway is genetically perturbed (*Watson et al., 2016*; *Bulcha et al., 2019*). B12-mechanism-I is elicited by two TFs, *nhr-10* and *nhr-68* that function as a persistence detector in a type I coherent feed-forward loop with AND-logic gate (*Bulcha et al., 2019*). We unraveled B12-mechanism-II in animals lacking *nhr-10* that cannot employ B12-mechanism-I (*Bulcha et al., 2019*). B12-mechanism-II targets the other vitamin B12-dependent metabolic pathway in *C. elegans*: the Met/SAM cycle. It is activated by low activity of this cycle that results in low levels of SAM and the activation of another NHR TF, *nhr-114*, which is not involved in B12-mechanism-I (*Bulcha et al., 2019*).

The discovery that Met/SAM cycle activity is sensed by a gene regulatory network resulting in the adjustment of Met/SAM cycle gene expression and in- and efflux modulation indicates that the animal strives to maintain the activity of this cycle in a tight metabolic regime to support development and homeostasis. Too little activity hampers the synthesis of phosphatidylcholine, an essential membrane component required for proliferation and growth, thereby inducing sterility and developmental delay. It is more difficult to assess why excessive Met/SAM cycle activity could be deleterious. One possibility may be the connection between Met/SAM cycle activity and folate status. When homocysteine levels are increased, the *metr-1* reaction is driven toward methionine production and consequently 5-methyltetrahydrofolate (5-meTHF) and folate are depleted (*Hasan et al., 2019*). The folate cycle is essential in maintaining nucleotide and NADPH pools, which are required for biomass and redox homeostasis (*Locasale, 2013*). SAM is not only required for the biosynthesis of phosphatidylcholine, but also it is the methyl donor for histone methylation (*Towbin et al., 2012*). High production of SAM could result in extensive histone modifications, which may result in aberrant overall gene expression (*Mentch et al., 2015*).

Previously, it has been shown that *sbp-1* is important for the expression of Met/SAM cycle genes (*Walker et al., 2011*). Interestingly, however, *sbp-1* is not specific to B12-mechanism-II as its perturbation also affects the response to propionate accumulation (*Bulcha et al., 2019*) (this study). Since *sbp-1* functions at a high level in the intestinal gene regulatory network, that is, it influences many gene expression programs, (*MacNeil et al., 2015*), it is likely that *sbp-1* responds to multiple different metabolic imbalances and activates each B12-mechanism. In support of this, *sbp-1* activates both *nhr-68* and *nhr-114*, which are important for B12-mechanism-I and B12-mechanism-II, respectively (*Walker et al., 2011*; *MacNeil et al., 2015*). Future studies will reveal the detailed wiring of *sbp-1*, *nhr-10*, *nhr-68*, and *nhr-114* into increasingly intricate gene regulatory networks.

The precise molecular mechanism by which the different NHRs mediate the two B12-mechanisms remains to be elucidated. We have previously shown that NHR-10 physically interacts with the *acdh-1* promoter (*Arda et al., 2010*; *Fuxman Bass et al., 2016*). However, we did not detect any insightful physical promoter-DNA or protein-protein interactions for either NHR-68 or NHR-114, and therefore it is not yet known which promoters or other TFs these TFs bind to (*Reece-Hoyes et al., 2013*; *Fuxman Bass et al., 2016*). It is also not yet clear how high propionate or low SAM levels are sensed. Since NHRs are liganded TFs, an interesting possibility would be that these metabolites function as ligands where propionate would interact with and activate NHR-10 and/or NHR-68, and SAM would interact with and inhibit NHR-114. Alternatively, the ratio between SAM and S-adenosylhomocysteine (SAH, *Figure 1A*) could be sensed by *nhr-114*, either directly or indirectly. Future detailed studies of propionate shunt and Met/SAM cycle gene promoters will likely provide insights into the molecular mechanisms governing both B12-mechanisms.

We have used the *Pacdh-1::GFP* dietary reporter as a crucial tool to elucidate B12-mechanism-I and B12-mechanism-II. While the reason for activating *acdh-1* expression by B12-mechanism-I is clear (it supports an alternate propionate breakdown mechanism), the reason for inducing this gene in response to Met/SAM cycle perturbations remains unclear. There are two possibilities: either perturbations in the Met/SAM cycle produce propionate which requires ACDH-1 to be degraded, or ACDH-1 catalyzes another reaction in addition to the conversion of propionyl-CoA into acrylyl-CoA. Future studies are required to determine if and how *acdh-1* functions in Met/SAM cycle metabolism.

Taken together, we have uncovered a second mechanism of gene regulation by vitamin B12 that ensures the flux of Met/SAM cycle metabolism to be in a tight, homeostatic regime.

# Materials and methods

**Key resources table**

| Reagent type (species) or resource | Designation | Source or reference | Identifiers | Additional information |
|---|---|---|---|---|
| Strain, strain background (*Caenorhabditis elegans*) | N2; wild type | Wormbase | RRID:WB-STRAIN: WBStrain00000001 | Laboratory reference strain/ wild type |
| Strain, strain background (*C. elegans*) | TM4695; Δ*nhr-10* | • Wormbase | Wormbase ID: WBVar00253059 | Genotype: *nhr-10(tm4695) III* |
| Strain, strain background (*C. elegans*) | VC1527; Δ*nhr-68* | Wormbase | RRID:WB-STRAIN: WBStrain00036665 | Genotype: *nhr-68(gk708) V* |
| Strain, strain background (*C. elegans*) | VL749; P*acdh-1::GFP* | Wormbase | RRID:WB-STRAIN: WBStrain00040155 | Genotype: *wwIs24[Pacdh-1::GFP;unc-119(+)]* |
| Strain, strain background (*C. elegans*) | VL868; '*Pacdh-1::GFP;Δnhr-10*' | PMID:23540701 | | Genotype: *wwIs24[Pacdh-1::GFP; unc-119(+)];nhr-10(tm4695)* |
| Strain, strain background (*C. elegans*) | VL1127; '*Pacdh-1::GFP;Δnhr-114*' | PMID:26430702 | | Genotype: *wwIs24[Pacdh-1::GFP;unc-119(+)]; nhr-114(gk849)* |
| Strain, strain background (*C. elegans*) | VL1102; '*Pacdh-1::GFP;Δmetr-1*' | PMID:23540702 | | Genotype: *wwIs24[Pacdh-1::GFP;unc-119(+)]; metr-1(ok521)* |
| Strain, strain background (*C. elegans*) | VL1115; '*Pacdh-1::GFP;Δsams-1*' | PMID:23540702 | | Genotype: *wwIs24[Pacdh-1::GFP;unc-119(+)]; sams-1(ok2946)* |
| Strain, strain background (*C. elegans*) | CB4856 | Wormbase | RRID:WB-STRAIN: WBStrain00004602 | Wild isolate |
| Strain, strain background (*C. elegans*) | VL1199; '*Pacdh-1::GFP; Δnhr-10;mthf-1(ww50)*' | This paper | | See Materials and methods, Genotype: *wwIs24[Pacdh-1::GFP;unc-119(+)];nhr-10(tm4695);mthf-1(ww50)* |
| Strain, strain background (*C. elegans*) | VL1200; '*Pacdh-1::GFP; Δnhr-10;sams-1(ww51)*' | This paper | | See Materials and methods, Genotype: *wwIs24[Pacdh-1::GFP;unc-119(+)]; nhr-10(tm4695); sams-1(ww51)* |
| Strain, strain background (*C. elegans*) | VL1201; '*Pacdh-1::GFP; Δnhr-10;metr-1(ww52)*' | This paper | | See Materials and methods, Genotype: *wwIs24[Pacdh-1::GFP;unc-119(+)]; nhr-10(tm4695); metr-1(ww52)* |

*Continued on next page*

*Continued*

| Reagent type (species) or resource | Designation | Source or reference | Identifiers | Additional information |
|---|---|---|---|---|
| Strain, strain background (*C. elegans*) | VL1205; '*Pacdh-1::GFP; Δnhr-10;mtrr-1(ww56)*' | This paper | | See Materials and methods, Genotype: *wwIs24[Pacdh-1::GFP;unc-119(+)];nhr-10(tm4695); mtrr-1(ww56)* |
| Sequence-based reagent | q_act-1_F | PMID:23540702 | qPCR primers | ctcttgcccc atcaaccatg |
| Sequence-based reagent | q_act-1_R | PMID:23540702 | qPCR primers | cttgcttgga gatccacatc |
| Sequence-based reagent | q_ama-1_F | PMID:23540702 | qPCR primers | agtgccgagat tgaaggaga |
| Sequence-based reagent | q_ama-1_R | PMID:23540702 | qPCR primers | gtattgcatgtt accttttcaacg |
| Sequence-based reagent | q_metr-1_F | This paper | qPCR primers | Used for qRT-PCR as described in methods, GGAGCAGCTAC TGGTAGAC |
| Sequence-based reagent | q_metr-1_R | This paper | qPCR primers | Used for qRT-PCR as described in methods, CACAGATGGCGA AATTGAGAG |
| Sequence-based reagent | q_mtrr-1_F | This paper | qPCR primers | Used for qRT-PCR as described in methods, TACGTTCTTC TCGGTCTCG |
| Sequence-based reagent | q_mtrr-1_R | This paper | qPCR primers | Used for qRT-PCR as described in methods, AGAGCTGTCA GTTGTTTGTC |
| Sequence-based reagent | q_sams-1_F | This paper | qPCR primers | Used for qRT-PCR as described in methods, ATTATCAAGG AGCTCGACCT |
| Sequence-based reagent | q_sams-1_R | This paper | qPCR primers | Used for qRT-PCR as described in methods, ATGGGAACTC AGAGTGACC |
| Sequence-based reagent | q_ahcy-1_F | This paper | qPCR primers | Used for qRT-PCR as described in methods, CGATTGCGAG ATTGACGTC |
| Sequence-based reagent | q_ahcy-1_R | This paper | qPCR primers | Used for qRT-PCR as described in methods, GTGTAACGGTC AACCTGTG |

*Continued on next page*

*Continued*

| Reagent type (species) or resource | Designation | Source or reference | Identifiers | Additional information |
|---|---|---|---|---|
| Sequence-based reagent | q_mel-32_F | This paper | qPCR primers | Used for qRT-PCR as described in methods, TGACTCATGGATTCTTCACCC |
| Sequence-based reagent | q_mel-32_R | This paper | qPCR primers | Used for qRT-PCR as described in methods, GATCAACCTTGTATGGAAGAGAC |
| Sequence-based reagent | q_mthf-1_F | This paper | qPCR primers | Used for qRT-PCR as described in methods, GTTGAGACCGATGAGAATGC |
| Sequence-based reagent | q_mthf-1_R | This paper | qPCR primers | Used for qRT-PCR as described in methods, TTCATAATGCTTTGGTGACCAG |
| Sequence-based reagent | q_pmt-2_F | This paper | qPCR primers | Used for qRT-PCR as described in methods, TTCATGTCGAAGTTTACCCA |
| Sequence-based reagent | q_pmt-2_R | This paper | qPCR primers | Used for qRT-PCR as described in methods, GTCCTTCTCGATGTATCCG |
| Sequence-based reagent | q_cbs-1_F | This paper | qPCR primers | Used for qRT-PCR as described in methods, GAAGCTAGAGTATCTCAATATTGCG |
| Sequence-based reagent | q_cbs-1_R | This paper | qPCR primers | Used for qRT-PCR as described in methods, CCAATCTCTTCAGCAAACTGG |
| Software, algorithm | CloudMap | PMID:23051646 | | |
| Software, algorithm | DolphinNext | PMID:32306927 | | |
| Software, algorithm | DESeq2 | PMID:25516281 | RRID:SCR_01568 | |
| Software, algorithm | WormFinder.m | This paper | | See methods, Available on github: https://github.com/shiaway/wormFinder/blob/master/wormFinder.m |
| Software, algorithm | GETprime | PMID:21917859 | | |
| Software, algorithm | fiji/imageJ | PMID:22743772 | RRID:SCR_002285 | |

### *C. elegans* strains

Animals were maintained on nematode growth media (NGM) as described (*Brenner, 1974*) with the following modifications. Soy peptone (Thomas Scientific) was used in place of bactopeptone and 0.64 nM vitamin B12 was added to maintain strains. N2 (Bristol) was used as the wild type strain. Animals were fed a diet of *E. coli* OP50 unless otherwise noted. The *wwIs24[Pacdh-1::GFP + unc-119 (+)]* (VL749) strain was described previously (*Arda et al., 2010*; *MacNeil et al., 2013*). *nhr-114 (gk849)*, *metr-1(ok521)*, and *sams-1(ok 2946)* were retrieved from the *C. elegans* Gene Knock-out Consortium (CGC), and *nhr-10(tm4695)* was obtained from the National Bioresource Project, Japan. All mutant strains were backcrossed three times with N2 wild type animals and crossed with VL749 prior to use in experiments. VL868 [*wwIs24[Pacdh-1::GFP + unc-119(+)];nhr-10(tm4695)*], VL1127 [*wwIs24[Pacdh-1::GFP + unc-119(+)];nhr-114(gk849)*], VL1102 [*wwIs24[Pacdh-1::GFP + unc-119(+)]; metr-1(ok521)*], and VL1115 [*wwIs24[Pacdh-1::GFP + unc-119(+)];sams-1(ok2946)*] are referred to in the text as Δnhr-10, Δnhr-114, Δmetr-1 and Δsams-1 respectively. *C. elegans* strain CB4856 (Hawaiian) strain was obtained from the CGC.

### Bacterial strains

*E. coli* OP50 and *E. coli* HT115 were obtained from the CGC and grown from single colony to saturation overnight in Luria-Bertani Broth (LB) at 37°C, shaking at 200 rpm. *E. coli* HT115 carrying RNAi plasmids were maintained on 50 µg/mL ampicillin.

### GFP intensity measurement using image analysis

Raw images were analyzed using Fiji/ImageJ (v1.53, *Schindelin et al., 2012*). Animals in brightfield images were outlined manually using the selection tool. Measurements of area, integrated density and mean gray value were redirected to the same animal in the corresponding fluorescent image. Several surrounding background measurements were also selected, and their mean gray values were averaged. Corrected total fluorescence was calculated by subtracting the product of the object's area and the mean gray value of the background from the object's integrated density as described previously (*McCloy et al., 2014*).

### EMS screen

The EMS mutagenesis protocol was adapted from *Jorgensen and Mango, 2002*. *Pacdh-1::GFP;nhr-10(tm4695)* animals were treated with 50 mM ethyl methanesulfonate (EMS, Sigma) for four hours and then washed five times with M9 buffer. Mutagenized animals were allowed to recover on NGM agar plates seeded with *E. coli* OP50, and 200 animals were picked and transferred to NGM agar plates containing 20 nM vitamin B12. F2 animals were screened for the presence of GFP. At least 8000 haploid genomes were screened, and 27 homozygous mutants were selected, 16 of which remained viable.

### Mutant mapping

Chromosome assignment was done by crossing EMS mutants into the CB4856 (Hawaiian) strain. Separate pools of GFP positive and GFP negative F2 animals were mapped using single-nucleotide polymorphisms as described (*Davis et al., 2005*).

### Whole genome sequencing

Mutant strains ww50, ww51, ww52, ww53, ww54, ww55, and ww56 were backcrossed four to five times to the VL868 [*wwIs24[Pacdh-1::GFP + unc-119(+)];nhr-10(tm4695)*] parental strain prior to sequencing. Genomic DNA was prepared by phenol-chloroform extraction and ethanol precipitation. Fragmentation was carried out on a Covaris sonicator E220 and 300–400 bp size fragments were collected using AMPure beads. Libraries were prepared and barcoded using the Kapa hyper prep kit (KK8500). Samples were sequenced at the core facility of the University of Massachusetts Medical school on an Illumina HiSeq4000 using 50 bp paired-end reads. After filtering out low-quality reads, 300 million reads were recovered resulting in an 18X average coverage of the genome. Reads were mapped to the *C. elegans* reference genome version WS220 and analyzed using the CloudMap pipeline (*Minevich et al., 2012*) where mismatches were compared to the parental strain

as well as to the other sequenced mutants. Variants with unique mismatches were validated by restriction fragment length polymorphism PCR (RFLP) and sanger sequencing.

## RNAi screen

RNAi screening was carried out as described (*Conte et al., 2015*). Briefly, RNAi clones were cultured in 96 well deep-well dishes in LB containing 50 µg/ml ampicillin and grown to log-phase at 37℃. Clone cultures were concentrated to 20-fold in M9 buffer and 10 µL was plated onto a well of a 96-well plate containing NGM agar with 2 mM Isopropyl β- d-1-thiogalactopyranoside (IPTG, Fisher Scientific). Plates were dried and stored at room temperature. The next day approximately 15–20 synchronized L1 animals per well were plated, followed by incubation at 20℃. Plates were screened 72 hr later. The metabolic gene RNAi screen using VL868 [*wwIs24[Pacdh-1::GFP + unc-119(+)];nhr-10 (tm4695)*] animals was performed twice. The TF RNAi screen using the Met/SAM cycle mutants generated by EMS was performed six times. All final hits were sequence-verified and retested on 35 mm NGM agar plates with approximately 200 animals per condition.

## Expression profiling by RNA-seq

Animals were treated with NaOH-buffered bleach, L1 arrested and plated onto NGM plates supplemented with 20 nM vitamin B12 and fed *E. coli* OP50. 400 late L4/early young adult animals were picked into M9 buffer, washed three times and flash frozen in liquid nitrogen. Total RNA was extracted using TRIzol (ThermoFisher), followed by DNase I (NEB) treatment and purified using the Direct-zol RNA mini-prep kit (Zymo research). RNA quality was verified by agarose gel electrophoresis and expression of known genes were measured via qRT-PCR for quality control. Two biological replicates were sequenced by BGI on the BGISEQ-500 next generation sequencer platform using 100 bp paired-end reads. A minimum of approximately 40 million reads was obtained per sample. Raw reads were processed on the DolphinNext RSEM v1.2.28 pipeline revision 7 (*Yukselen et al., 2020*). In brief, the reads were mapped by bowtie2 to genome version c_elegans.PRJNA13758 (WormBase WS271), and then passed to RSEM for estimation of TPM and read counts. Default parameters were used for both bowtie2 and RSEM.

For later RNA-seq experiments we have developed a more cost-effective, in-house method for RNA-sequencing. Briefly, multiplexed libraries were prepared using Cel-seq2 (*Hashimshony et al., 2016*). Two biological replicates were sequenced with a NextSeq 500/550 High Output Kit v2.5 (75 Cycles) on a Nextseq500 sequencer. Paired end sequencing was performed; 13 cycles for read 1, six cycles for the illumina index and 73 cycles for read 2. Approximately 12 million reads per sample was achieved.

The libraries were first demultiplexed by a homemade python script, and adapter sequences were trimmed using trimmomatic-0.32 by recognizing polyA and barcode sequences. Then, the alignment to the reference genome was performed by STAR with the parameters '—runThreadN 4 – `alignIntronMax` 25000 `–outFilterIntronMotifs` RemoveNoncanonicalUnannotated'. Features were counted by ESAT (*Derr et al., 2016*) with the parameters '-task score3p -wLen 100 -wOlap 50 -wExt 1000 -sigTest. 01 -multimap normal -scPrep -umiMin 1'. Features in 'c_elegans.PRJNA13758. WS271.canonical_geneset.gtf' were used as the annotation table input for ESAT, but pseudogenes were discarded. The read counts for each gene were used in differential expression analysis by DEseq2 package in R 3.6.3 (*Love et al., 2014*). A fold change cut off of greater than 1.5 and *P* adjusted value cut off of less than 0.01 was used. All the processing procedures were done in a homemade DolphinNext pipeline.

The RNA-sequencing data files were deposited in the NCBI Gene Expression Omnibus (GEO) under the following accession numbers:

- *Bulcha et al., 2019*: GSE123507
- This study: GSE151848

## Expression profiling by qRT-PCR

Animals were grown and harvested, and RNA was extracted as described for the RNA-seq experiments. qRT-PCR was performed as described previously (*Bulcha et al., 2019*). cDNA was reverse transcribed from total RNA using oligo(dT) 12–18 primer (Invitrogen) and Mu-MLV Reverse Transcriptase (NEB). qPCR primers were designed using the GETprime database (*Gubelmann et al.,*

*2011*). qPCR reactions were carried out in technical triplicate using the StepOnePlus Real-Time PCR system (Applied Biosystems) and Fast Sybr Green Master Mix (ThermoFisher Scientific). Relative mRNA transcript abundance was calculated using the ΔΔCT method (*Schmittgen and Livak, 2008*) and normalized to the geometric mean of *ama-1* and *act-1* levels.

### Body size measurement

Approximately 100 synchronized L1 animals were plated across four wells of a 48-well plate per condition. L4 animals were collected and washed three times in 0.03% sodium azide and transferred to a 96-well plate. Excess liquid was removed, and plates were rested for an hour to allow animals to settle and straighten. Pictures were taken using an Evos Cell Imaging System microscope and image processing was done using a MATLAB (MathWorks) script named 'wormFinder.m' written in-house and made available at the following link: https://github.com/shiaway/wormFinder/blob/master/wormFinder.m (*Ponomarova and Giese, 2020*)

### Gas chromatography-mass spectrometry

For *Figure 5A*, gravid adults were harvested from liquid S media cultures supplemented with or without 64 nM vitamin B12 and fed concentrated *E. coli* OP50. For *Figure 5E*, gravid adults were harvested from NGM agar plates treated with 64 nM vitamin B12 and seeded with *E. coli* OP50. Animals were washed in 0.9% saline until the solution was clear and then twice more (3–6 times total). Metabolites were extracted and analyzed as described previously (*Na et al., 2018*). Briefly, 50 μL of a semi-soft pellet of animals was transferred to a 2 mL FastPrep tube (MP Biomedicals) and flash frozen with liquid nitrogen. Metabolites were extracted in 80% cold methanol. Acid-washed micro glass beads (Sigma) and a FastPrep-24 5G homogenizer (MP Biomedicals) were used to disrupt animal bodies. After settling, supernatant was transferred to glass vials (Sigma) and dried by speed-vac overnight. MeOX-MSTFA derivatized samples were analyzed on an Agilent 7890B/5977B single quadrupole GC-MS equipped with an HP-5ms Ultra Inert capillary column (30 m × 0.25 mm×0.25 μm) using the same method as described (*Na et al., 2018*).

## Acknowledgements

We thank Amy Walker, Jenny Benanti, Mark Alkema, Job Dekker, and members of the Walhout lab for discussion and critical reading of the manuscript. We thank Amy Walker for the *pmt-2* RNAi clone. This work was supported by a grant from the National Institutes of Health (DK068429) to AJMW. Some bacterial and nematode strains used in this work were provided by the CGC, which is funded by the NIH Office or Research Infrastructure Programs (P40 OD010440).

## Additional information

### Funding

| Funder | Grant reference number | Author |
| --- | --- | --- |
| National Institutes of Health | DK068429 | Albertha JM Walhout |

The funders had no role in study design, data collection and interpretation, or the decision to submit the work for publication.

### Author contributions

Gabrielle E Giese, Conceptualization, Data curation, Software, Formal analysis, Supervision, Funding acquisition, Investigation, Visualization, Methodology, Writing - original draft, Project administration, Writing - review and editing, GEG and AJMW conceived the study, GEG performed all experiments with technical help from MW, GEG and AJMW wrote the manuscript; Melissa D Walker, Investigation, Writing - review and editing, Provided technical help with experiments; Olga Ponomarova, Conceptualization, Data curation, Software, Formal analysis, Investigation, Visualization, Methodology, Writing - original draft, Writing - review and editing, OP wrote the body size measurement MATLAB script and performed some GC-MS experiments; Hefei Zhang, Investigation, Methodology, Writing - review and editing, HZ and XL performed the RNA-seq experiment and analysis from

samples generated by GEG in Figures 4 and 5; Xuhang Li, Software, Formal analysis, Investigation, Methodology, Writing - review and editing, HZ and XL performed the RNA-seq experiment and analysis from samples generated by GEG in Figures 4 and 5; Gregory Minevich, Software, Formal analysis, Investigation, Methodology, Writing - review and editing, GM ran the CloudMap utility to analyze the whole genome sequencing data; Albertha JM Walhout, Conceptualization, Formal analysis, Supervision, Funding acquisition, Writing - original draft, Project administration, Writing - review and editing, GEG and AJMW conceived the study, GEG and AJMW wrote the manuscript

## Author ORCIDs

Gabrielle E Giese (iD) https://orcid.org/0000-0002-9338-2049
Albertha JM Walhout (iD) https://orcid.org/0000-0001-5587-3608

## Decision letter and Author response

Decision letter https://doi.org/10.7554/eLife.60259.sa1
Author response https://doi.org/10.7554/eLife.60259.sa2

---

# Additional files

## Supplementary files

• Supplementary file 1. mRNA fold change and p adjusted values of genes involved in the propionate shunt, Met/SAM cycle and phospholipid metabolism in the following conditions. Wild type on vitamin B12 vs.wild type on untreated; wild type on vitamin B12 and propionate vs. wild type on vitamin B12; Δnhr-10 untreated vs. wild type untreated; Δnhr-10 on vitamin B12 vs. Δnhr-10 untreated; Δnhr-68 untreated vs. wild type untreated; Δnhr-68 on vitamin B12 vs. Δnhr-68 untreated. This table is a subset of data published in *Bulcha et al., 2019* (GSE123507). Related to *Figures 1*, *3*, *5* and *6*.

• Supplementary file 2. Genes with an mRNA fold change greater or equal to 1.5 and *a p* adjusted value less than 0.01 in Met/SAM cycle mutants vs.the parental strain *Pacdh-1::GFP;Δnhr-10.* Genes that change in all four mutant strains are shown in tab 1, while all genes regardless of p adjusted value are shown in tab 2. Related to *Figures 3* and *6*.

• Supplementary file 3. mRNA fold change and p adjusted values of all detected genes in the following conditions. *Pacdh-1::GFP* on *nhr-114* RNAi vs. vector; *Pacdh-1::GFP;Δmetr-1* on vector vs. *Pacdh-1::GFP* on vector; *Pacdh-1::GFP;Δmetr-1* on *nhr-114* RNAi vs. *Pacdh-1::GFP* on vector; *Pacdh-1::GFP;Δmetr-1* on *nhr-114* RNAi vs. *Pacdh-1::GFP;Δmetr-1* on vector; *Pacdh-1::GFP;Δnhr-10;metr-1 (ww52)* on vector vs. *Pacdh-1::GFP* on vector; *Pacdh-1::GFP;Δnhr-10;metr-1(ww52)* on *nhr-114* RNAi vs. *Pacdh-1::GFP* on vector; *Pacdh-1::GFP;Δnhr-10;metr-1(ww52)* on *nhr-114* RNAi vs. *Pacdh-1:: GFP;Δnhr-10;metr-1(ww52)* on vector. Related to *Figure 4*.

• Supplementary file 4. mRNA fold change and p adjusted values of all detected genes in the following conditions. *Pacdh-1::GFP* on methionine vs. *Pacdh-1::GFP* untreated; *Pacdh-1::GFP;Δnhr-114* untreated vs. *Pacdh-1::GFP* untreated; *Pacdh-1::GFP;Δnhr-114* on methionine vs. *Pacdh-1::GFP* on methionine; *Pacdh-1::GFP;Δnhr-114* on methionine vs. *Pacdh-1::GFP;Δnhr-114* untreated. Related to *Figures 5* and *6*.

• Transparent reporting form

## Data availability

Sequencing data have been deposited in GEO under accession codes GSE123507 and GSE151848.

The following dataset was generated:

| Author(s) | Year | Dataset title | Dataset URL | Database and Identifier |
|---|---|---|---|---|
| Giese GE | 2020 | C. elegans methionine/S-adenosylmethionine cycle activity is sensed and adjusted by a nuclear hormone receptor | http://www.ncbi.nlm.nih.gov/geo/query/acc.cgi?acc=GSE151848 | NCBI Gene Expression Omnibus, GSE151848 |

The following previously published dataset was used:

| Author(s) | Year | Dataset title | Dataset URL | Database and Identifier |
|---|---|---|---|---|
| Bulcha JT, Walhout AJ | 2019 | A persistence detector for metabolic network rewiring in an animal | https://www.ncbi.nlm.nih.gov/geo/query/acc.cgi?acc=GSE123507 | NCBI Gene Expression Omnibus, GSE123507 |

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
