## [Decision Letter]

Thank you for submitting your article "*C. elegans* methionine/S-adenosylmethionine cycle activity is sensed and adjusted by a nuclear hormone receptor" for consideration by *eLife*. Your article has been reviewed by three peer reviewers, one of whom is a member of our Board of Reviewing Editors, and the evaluation has been overseen by Naama Barkai as the Senior Editor. The reviewers have opted to remain anonymous.

All reviewers agree that this is a very nice paper providing a number of novel and interesting insights into the transcriptional control of metabolism. As you will see below, there are a number of relatively small issues that related to data analysis and clarity of presentation which should be all very straightforward to address. We do not expect that a properly revised manuscript will need to be sent out for review again.

Reviewer #1:

In this study, Giese et al. have discovered a novel autoregulatory mechanism termed as 'vitamin B12-mechanism-II' that is activated in response to vitamin B12 deficiency. Vitamin B12 is required for the synthesis of S-adenosylmethionine (SAM) in Met/SAM cycle arm of one-carbon metabolism. When SAM synthesis is low due to vitamin B12 deficiency or genetic perturbations of Met/SAM cycle enzymes, this results in upregulation of multiple Met/SAM cycle genes by the nuclear hormone receptor NHR-114. In addition, low levels of SAM due to vitamin B12 deficiency results in activation of the vitamin B12 transporter *pmp-5* and increased influx and reduced efflux of SAM precursors in the Met/SAM cycle in an *nhr-114*-dependent manner. The authors further speculate that reduced SAM level acts as the direct sensor for the activation of vitamin B12-mechanism-II.

This study makes a major conceptual contribution to our understanding of vitamin B12-dependent metabolism and the gene regulatory networks that are activated in response to B12 deficiency. The experiments are carefully designed to unravel the epistatic nature of interactions in this complex metabolic signaling pathway.

There are only two concerns regarding interpretation of the data shown in this manuscript: (i) many crucial interpretations are based on a *Pacdh-1::GFP* reporter (which is fine), but the GFP intensity is not quantified, and (ii) the genetic interactions proposed in the model are primarily derived from gene expression changes measured via RNA-seq for only two biological replicates. In some cases, the effect sizes are quite large, but statistical significance should be reported especially for two-way interaction (epistatic) data.

1) Subsection “Low dietary vitamin B12 activates two transcriptional mechanisms”: Given the data shown in Figure 1B, it cannot be interpreted conclusively that the activation of *acdh-1* expression in response to accumulating propionate is completely dependent on *nhr-10*. This is because the experiment lacks the propionate only control. The authors should quantify the GFP signal intensity change in wild-type and *nhr-10* mutant worms after addition of propionate alone. Alternatively, the authors can rephrase this sentence in the context of propionate accumulation in the presence of exogenously added vitamin B12 + propionate.

2) Subsection “Low dietary vitamin B12 activates two transcriptional mechanisms”: The GFP expression levels should be quantified in Figure 1B. It is not apparent that the GFP signal intensity in untreated *nhr-10* mutant worms is lower than that in untreated wild-type worms, though the tissue-specific localization of the GFP signal seems different in the two strains.

3) Subsection “*nhr-114* mediates Met/SAM cycle activation in response to low Met/SAM cycle flux”: The data shown in Figure 4D are central to the model proposed in this paper. Hence, these results need to be backed by statistical testing. The authors can perform two-way ANOVA (or two-way repeated measures ANOVA) on the RNA-seq data for these six Met/SAM cycle genes. If statistical significance is not obtained from the two replicates used in RNA-seq, the authors should validate the expression changes of these six genes across the different conditions using qRT-PCR for additional biological replicates.

4) “However, while choline supplementation repressed GFP expression in *sams-1*(*ww51*) mutants, methionine supplementation did not (Figure 5D)”: This claim can be made only if the GFP signal intensity is quantified in the panels of Figure 5D. Based on the representative image, there seems to be a reduction in GFP intensity in the *Pacdh-1::GFP*; ∆*nhr-10*; *sams-1*(*ww51*) strain after methionine supplementation, which the author also discuss later in the same section.

5) “The expression of *pmt-2* is repressed by vitamin B12, but not activated by propionate, is not under the control of *nhr-10* or *nhr-68*, and is activated in Met/SAM cycle mutants (Bulcha et al., 2019) (Figure 5F; Supplementary file 2)”: Some statistical testing is required to make this claim (similar to the statistical comparisons performed in Figure 5A and E). If statistical significance is not obtained from the two replicates used in RNA-seq, the authors should validate the expression changes using qRT-PCR for additional biological replicates.

6) “Methionine and choline supplementation suppress B12-Mechanism-II”: *pmt-2* RNAi decreases GFP expression not only in *Pacdh-1::GFP*; ∆*nhr-10* animals, but also induces a very strong reduction in GFP expression in *Pacdh-1::GFP* wild-type animals. What might be the potential cause of this? In other words, how does depletion of *pmt-2* prevent *acdh-1* activation via B12-mechanism-I? Does *pmt-2* RNAi prevent propionate accumulation in wild-type worms?

7) Subsection “Methionine and choline supplementation suppress B12-Mechanism-II”, last paragraph: These results should be backed by statistical testing. The authors can perform two-way ANOVA (or two-way repeated measures ANOVA) on the RNA-seq data for these six Met/SAM cycle genes.

8) Figure 4D versus Figure 5H: It is very interesting that *nhr-114* depletion results in a major reduction in the expression of all six Met/SAM cycle genes in the absence of vitamin B12 (Figure 5H), but it has no effect on the expression of these genes in the presence of exogenously supplemented vitamin B12 (Figure 4D). This finding is consistent with the model proposed in the paper and the authors should discuss this in the manuscript.

9) Figure 6A-D: Statistical tests should be performed for the differences in gene expression across conditions and p values should be reported in the figure.

10) Subsection “B12-mechanism-II actives influx and represses efflux of the Met/SAM cycle”: The authors should discuss why depletion of *nhr-10* and *nhr-68*, regulators of vitamin B12-mechanism-I, also results in reduced *pmp-5* expression.

11) Subsection “B12-mechanism-II actives influx and represses efflux of the Met/SAM cycle”: p value for statistical test performed should be depicted on the figure (*nhr-114* expression in wild-type vs. *nhr-68* mutant).

Reviewer #2:

Previously this group identified a new shunt pathway for propionate breakdown, which (in contrast to the canonical pathway) does not involve a vitamin B12-dependent step and which is transcriptionally activated by low vitamin B12 levels. Accumulating propionate induces the first gene in this new shunt pathway, *acdh-1*, and this induction is completely dependent on *nhr-10*. In this manuscript, the authors find that even in the absence of *nhr-10*, *acdh-1* is induced at a low level, indicating a propionate-independent mechanism to induce *acdh-1*. The propionate-independent mechanism is induced by low vitamin B12, which leads to low SAM and requires another NHR, *nhr-114*. This pathway induces expression of Met/SAM cycle genes as well as other genes that help to compensate for the low vitamin B12. This paper illustrates the power of *C. elegans* to uncover the transcriptional mechanisms that control basic metabolic pathways and are still not well understood. The experiments and analysis in this paper are well done.

It would be helpful if the second and third paragraph in the Introduction referenced and mapped to the overview in Figure 1A. MS and MUT should be indicated in Figure 1A.

Figure 7 is confusing. What do the different colors mean? Why are there both positive and negative signals between B12 and the Met/SAM cycle? A more detailed figure legend would certainly help. Ideally, the legend would briefly review the meaning/evidence for each of the positive/negative signals.

I would like the authors to comment on the findings of Gracida and Eckmann that Trp, but not other amino acids, could rescue the sterility of *nhr-114* mutants, while the authors found that methionine (and choline and B12) could rescue.

Reviewer #3:

The Walhout lab had previously demonstrated that dietary deficiency in vitamin B12 results in compensatory transcriptional responses. They had found that propionate accumulation due to vitamin B12 deficiency activates expression of set of genes that encode for components of a propionate shunt system. These activations are dependent on *nhr-10*. They refer to this pathway as "B12-mechanism-I". Their discovery was aided by use of a reporter, *Pacdh-1::GFP*, whose expression is upregulated upon vitamin B12 deficiency. However, *Pacdh-1::GFP* is not fully shut down in *nhr-10* mutants when subjected to low B12 levels The current paper details the discovery of what they refer to as "B12-mechanism-II'. Using a combination of forward and reverse genetic screens, gene expression studies, metabolite measurements, and metabolite add back experiments, the authors show that low concentrations of SAM underlie this second mechanism and that the noted gene upregulations are dependent on *nhr-114*.

Overall, this is an excellent paper and well suited for publication in *eLife*. The experiments are logical and well executed. The data are sufficiently robust and well controlled. The conclusions are supported by corroborative pieces of evidence. The authors should be commended for their clear writing, especially since the pathways are hard to keep track of and the discoveries depend on complex combinations of conditions. I also thought that the authors did an excellent job pointing out unanswered question in the Discussion section of the manuscript.

I am happy to say that I have no major concerns about the manuscript.

---

## [Author Response]

Reviewer #1:[…]1) Subsection “Low dietary vitamin B12 activates two transcriptional mechanisms”: Given the data shown in Figure 1B, it cannot be interpreted conclusively that the activation of acdh-1 expression in response to accumulating propionate is completely dependent on nhr-10. This is because the experiment lacks the propionate only control. The authors should quantify the GFP signal intensity change in wild-type and nhr-10 mutant worms after addition of propionate alone. Alternatively, the authors can rephrase this sentence in the context of propionate accumulation in the presence of exogenously added vitamin B12 + propionate.

We know that *Pacdh-1::GFP* is activated by multiple metabolic perturbations (Bulcha et al., 2019; this study; unpublished data). Therefore, we thought the cleanest evidence for the requirement of *nhr-10* in the propionate response would be to first repress the reporter by adding vitamin B12 and then adding back in propionate.

However, we do have more images from the same experiment of wild type and ∆*nhr-10* mutant animals when propionate is supplemented without vitamin B12. We have included these as a supplementary figure in the revised manuscript (Figure 1—figure supplement 1). We respectfully argue that the visual assessment of GFP levels should be clear in this experiment. We also note that the observations with the *Pacdh-1::GFP* reporter have been corroborated throughout this study and our previous publication (Bulcha et al., 2019).

2) Subsection “Low dietary vitamin B12 activates two transcriptional mechanisms”: The GFP expression levels should be quantified in Figure 1B. It is not apparent that the GFP signal intensity in untreated nhr-10 mutant worms is lower than that in untreated wild-type worms, though the tissue-specific localization of the GFP signal seems different in the two strains.

We thank the reviewer for this comment. The overall intensity of GFP is reduced in *∆nhr-10* mutant animals when comparing the entire intestine, which is due to a dramatic reduction in the anterior part of the intestine. We have revised the manuscript to make this point: “Interestingly, we found that while GFP levels are reduced in the anterior intestine, there is still remaining GFP expression in the posterior intestine in *Pacdh-1::GFP* transgenic animals lacking *nhr-10*”.

3) Subsection “nhr-114 mediates Met/SAM cycle activation in response to low Met/SAM cycle flux”: The data shown in Figure 4D are central to the model proposed in this paper. Hence, these results need to be backed by statistical testing. The authors can perform two-way ANOVA (or two-way repeated measures ANOVA) on the RNA-seq data for these six Met/SAM cycle genes. If statistical significance is not obtained from the two replicates used in RNA-seq, the authors should validate the expression changes of these six genes across the different conditions using qRT-PCR for additional biological replicates.

See above.

4) “However, while choline supplementation repressed GFP expression in sams-1(ww51) mutants, methionine supplementation did not (Figure 5D)”: This claim can be made only if the GFP signal intensity is quantified in the panels of Figure 5D. Based on the representative image, there seems to be a reduction in GFP intensity in the Pacdh-1::GFP; ∆nhr-10; sams-1(ww51) strain after methionine supplementation, which the author also discuss later in the same section.

See above.

5) “The expression of pmt-2 is repressed by vitamin B12, but not activated by propionate, is not under the control of nhr-10 or nhr-68, and is activated in Met/SAM cycle mutants (Bulcha et al., 2019) (Figure 5F; Supplementary file 2)”: Some statistical testing is required to make this claim (similar to the statistical comparisons performed in Figure 5A and E). If statistical significance is not obtained from the two replicates used in RNA-seq, the authors should validate the expression changes using qRT-PCR for additional biological replicates.

See above.

6) “Methionine and choline supplementation suppress B12-Mechanism-II”: pmt-2 RNAi decreases GFP expression not only in Pacdh-1::GFP; ∆nhr-10 animals, but also induces a very strong reduction in GFP expression in Pacdh-1::GFP wild-type animals. What might be the potential cause of this? In other words, how does depletion of pmt-2 prevent acdh-1 activation via B12-mechanism-I? Does pmt-2 RNAi prevent propionate accumulation in wild-type worms?

Due to both B12-mechanism-I and B12-mechanism-II being intact in wild type animals it is difficult to uncouple the effects of each mechanism by *pmt-2* knock down on *Pacdh-1::GFP*. It is also possible, however, that there is some connection between phosphatidylcholine production and propionate. This potential connection remains to be explored.

7) Subsection “Methionine and choline supplementation suppress B12-Mechanism-II”, last paragraph: These results should be backed by statistical testing. The authors can perform two-way ANOVA (or two-way repeated measures ANOVA) on the RNA-seq data for these six Met/SAM cycle genes.

See above.

8) Figure 4D versus Figure 5H: It is very interesting that nhr-114 depletion results in a major reduction in the expression of all six Met/SAM cycle genes in the absence of vitamin B12 (Figure 5H), but it has no effect on the expression of these genes in the presence of exogenously supplemented vitamin B12 (Figure 4D). This finding is consistent with the model proposed in the paper and the authors should discuss this in the manuscript.

We agree this is a very important point which we think is conveyed by the following sentence in the manuscript:

“Interestingly, in the presence of vitamin B12, basal Met/SAM cycle gene expression is not affected by *nhr-114* RNAi.”

9) Figure 6A-D: Statistical tests should be performed for the differences in gene expression across conditions and p values should be reported in the figure.

See above.

10) Subsection “B12-mechanism-II actives influx and represses efflux of the Met/SAM cycle”: The authors should discuss why depletion of nhr-10 and nhr-68, regulators of vitamin B12-mechanism-I, also results in reduced pmp-5 expression.

We thank the reviewer for pointing this out. Based on homology with human *ABCD4*, we predict that PMP-5 transports vitamin B12 from the lysosome to the cytosol. Therefore, it’s depletion results in reduced availability of vitamin B12 and the activation of both vitamin B12-mechanism-I and mechanism-II. We have added the following to the revised manuscript:

“As a putative transporter of vitamin B12 from the lysosome to the cytosol, *pmp-5* is also important for vitamin B12-mechanism-I. Indeed, *pmp-5* is upregulated in response to propionate supplementation, and is also regulated by *nhr-10* and *nhr-68*”.

11) Subsection “B12-mechanism-II actives influx and represses efflux of the Met/SAM cycle”: p value for statistical test performed should be depicted on the figure (nhr-114 expression in wild-type vs. nhr-68 mutant).

See above.

Reviewer #2:[…] It would be helpful if the second and third paragraph in the Introduction referenced and mapped to the overview in Figure 1A. MS and MUT should be indicated in Figure 1A.

We agree and have included a reference to Figure 1A, we have also included *MS* and *MUT* in the revised figure.

Figure 7 is confusing. What do the different colors mean? Why are there both positive and negative signals between B12 and the Met/SAM cycle? A more detailed figure legend would certainly help. Ideally, the legend would briefly review the meaning/evidence for each of the positive/negative signals.

As also suggested by Reviewer #1, we have included a more detailed legend for Figure 7.

I would like the authors to comment on the findings of Gracida and Eckmann that Trp, but not other amino acids, could rescue the sterility of nhr-114 mutants, while the authors found that methionine (and choline and B12) could rescue.

Respectfully, it is difficult to know why Gracida *et al.* did not find that methionine could rescue *nhr-114* deficient animals. It could be methodological; for instance, Gracida *et al.* added bulk amino acids instead of adding amino acids individually. They also didn’t test different concentrations. It is possible that the effect of methionine was masked by other metabolites and/or the concentration was not appropriate for a noticeable rescue. To illustrate this, we have revised the text as follows:

“Although Gracida and Eckmann did not find that methionine rescued *nhr-114* RNAi knock down animals, it is possible this was due to methodological differences. For example, they added bulk L-amino acids and at only one concentration. The actual concentration of ingested methionine might have been masked by other amino acids or was simply too low to provide an observable rescue”.